# Meta-amplified dark-field interferometric scattering microscopy

Hongki Lee [1,6], Junxiang Zhao[1,6], Pan Hu[2], Zhaoyu Nie[3], Ming Lei[1], Guanghao Chen[1], Soojeong Baek[1], Qianyi Wu[1], Li Chen[1], Ang Li[1], Mojie Luo[1], Shizhen Emily Wang[4], Sui Yang [5], Wei Wu [2] & Zhaowei Liu [1] ✉

Label-free optical detection of nanometer-scale bioparticles is highly desirable for noninvasive biological studies but challenging due to the weak scattering signals that are difficult to distinguish from the illumination background. Interferometric scattering microscopy (iSCAT) has enabled high-sensitivity imaging by detecting the interference between the particle's scattered light and a reference beam. However, enhancing the detection sensitivity and the image contrast for small particles continues to be a challenge in iSCAT. Here, we introduce meta-amplified dark-field interferometric scattering microscopy (MAD-iSCAT), which leverages a plasmonic metasurface to drastically enhance nanoparticle detection sensitivity in iSCAT. By employing a metasurface comprising sub-diffraction plasmonic meta-atom arrays, MAD-iSCAT generates bright radiation modes that intensely scatter light toward the far field in the presence of a detection nanoparticle, substantially amplifying the sensitivity. In the absence of a nanoparticle, the metasurface produces minimal background due to the dark collective mode, resulting in improved image contrast. We present a theoretical analysis of amplified interferometric imaging using designed metasurfaces and experimentally demonstrate enhancements in contrast and signal-to-noise ratio for detecting dielectric nanoparticles, exosomes, and proteins. Our approach offers broad applications in label-free biosensing and optical mass spectrometry, enabling significantly improved throughput and sensitivity.

The identification and analysis of biomolecular activities and interactions are crucial for understanding physiological and pathological processes. Direct observation of nanometer-scale bioparticles using light microscopy without fluorescence labeling enables noninvasive investigation with excellent biocompatibility[1]. However, as the light scattering cross-section ($\sigma_{sc}$) scales with the sixth power of the particle's diameter, improving the detection sensitivity and contrast between the weak signal and the background noise has been a key challenge in optical nanoparticle imaging[2]. Interferometric imaging addresses this challenge by leveraging the interference of scattered light from nanoscale objects with reference light, yielding label-free images with high sensitivity. Recent advancements in interferometric scattering microscopy (iSCAT) have demonstrated the exceptional sensitivity in quantifying small particles, with applications ranging from sensing and tracking nanoparticles[3–6], proteins[7,8], and viruses[9], as well as enabling label-free intracellular imaging[10–12] and mass spectroscopy[13–15].

[1]Department of Electrical and Computer Engineering University of California, San Diego, CA, USA. [2]Ming Hsieh Department of Electrical Engineering University of Southern California, Los Angeles, CA, USA. [3]Department of Mechanical Engineering University of California, Berkeley, CA, USA. [4]Department of Pathology University of California, San Diego, CA, USA. [5]Materials Science and Engineering School for Engineering of Matter Transport and Energy Arizona State University, Tempe, AZ, USA. [6]These authors contributed equally: Hongki Lee, Junxiang Zhao. ✉e-mail: zhaowei@ucsd.edu

In iSCAT, the image contrast $C$ can be written as

$$C = 2\frac{\mathbf{E}_s}{\mathbf{E}_r}\cos(\alpha), \tag{1}$$

where $\mathbf{E}_s$ is the scattered electric field from the detection particle, $\mathbf{E}_r$ is the electric field of the reference light, and $\alpha$ is the phase difference between $\mathbf{E}_s$ and $\mathbf{E}_r$[16]. It can be readily seen that $\mathbf{E}_r$ needs to be minimized to ensure a sufficiently large image contrast for practical detection dynamic range. However, decreasing $\mathbf{E}_r$ is accompanied by a reduction in total detected photon counts. For detecting nanometer-scale dielectric objects, such a reduction leads to a low signal-to-noise ratio (SNR) or indistinguishable signals due to the shot-noise limit[13]. In a typical iSCAT, a partial reflector is deployed to reduce the reference beam intensity before detection[10,12,14,17]. Recently, substantial efforts have been dedicated to engineering $\mathbf{E}_s$ and $\mathbf{E}_r$ to attain optimal detection sensitivity, including pupil engineering[17–19] and optimizing the interfacing structures such as dielectric layers[18], plasmonic nanostructures[20], photonic resonators[21], and nanofluidic channels[22]. To obtain high SNR and large detection dynamic range, differential analysis and other algorithmic approaches have been developed for single protein detection and detailed subcellular imaging[9,12–14,23].

Meanwhile, the field of plasmonics has been developed over the past two decades, enabling numerous applications. Plasmonic nanostructures can produce highly intensified electric fields within deep-subwavelength regions, enhancing light interaction with adjacent objects. Such localized optical modes are extremely sensitive to surrounding environments and have found wide applications in biosensing technology[24,25], optical microscopy[26,27], photocatalyst[28], and more[29]. Recently, metasurfaces comprising subwavelength arrays of plasmonic nanostructures have emerged as a prominent research area due to their unique optical properties resulting from their collective response[30,31]. They have been adapted to many optical imaging techniques such as holography[32,33], edge detection[34,35], and others[36,37], demonstrating substantial improvements compared to conventional optics alone.

Here, we propose an interferometric imaging technique, called meta-amplified dark-field interferometric scattering microscopy (MAD-iSCAT), leveraging the benefits of plasmonic metasurfaces and dark-field interferometric imaging. MAD-iSCAT employs a plasmonic metasurface as the reference substrate instead of a conventional glass substrate, providing a uniform dark background. As a nanoparticle approaches, the plasmonic metasurface transitions to a radiation mode, amplifying the scattered signal and thus enhancing the nanoparticle detection. This approach can improve the contrast and sensitivity of traditional iSCAT.

## Results

### Principles of contrast and SNR amplification in interferometric images

Conventional iSCAT typically employs a bare glass substrate, where the reflected light generates an interference image of the scatterers to be detected. Similar label-free imaging methods have been introduced using dark-field illumination, which offers excellent background attenuation[38–41]. It has been shown that interferometric dark-field images are also generated through the interference between scattered lights from the object and surface roughness (static background)[39,40,42]. Fig.1a,b illustrates a schematic of MAD-iSCAT using a plasmonic metasurface with a conventional dark-field illumination. When a scatterer to be detected is located at $(0, 0, z_s)$ on the metasurface comprising an array of meta-atom elements, the detected signal is composed of the superposition of propagating electromagnetic (EM) fields from the meta-atom elements at $(x_{ij}, y_{ij}, -z_a)$ ($-\infty \leq i \leq \infty$ and $-\infty \leq$

$j \leq \infty$) and the scatterer as

$$
\begin{aligned}
P &\propto \left|\sum_{(i,j)}\mathbf{E}_{ij} + \mathbf{E}_s\right|^2 \\
&= \left|\sum_{(i,j)}\mathbf{E}_{ij}\right|^2 + 2\sum_{(i,j)}\left|\mathbf{E}_{ij}\right|\left|\mathbf{E}_s\right|\cos\left(\alpha_{ij}\right) + \left|\mathbf{E}_s\right|^2
\end{aligned}
\tag{2}
$$

where $\mathbf{E}_{ij}$ and $\mathbf{E}_s$ are the electric field distribution generated by the meta-atom elements at $(x_{ij}, y_{ij}, -z_a)$ and the scatterer at $(0, 0, z_s)$, respectively. $\alpha_{ij}$ denotes the difference in radiation phases between the meta-atom elements and the scatterer. For deep-subwavelength scatterers, the detected scattering intensity $|\mathbf{E}_s|^2$ scales with the sixth power of the particle size in conventional dark-field imaging (Fig. 1c). In previous and dark-field iSCAT schemes, $\mathbf{E}_{ij}$ terms in Eq. (2) are regarded as a static random background. Especially, $\left|\sum_{(i,j)}\mathbf{E}_{ij}\right|^2$ is not involved in the scattered light detection and is removed by subtracting consecutive frames.

In this study, we introduce a metasurface comprising subwavelength meta-atom arrays to improve the detection sensitivity in iSCAT. Each metallic meta-atom possesses a large $\sigma_{sc}$ compared to the dielectric nanoparticle (Fig. 1d). Without a nanoscatterer, the meta-atom array behaves as a sub-diffraction grating, primarily reflecting incident light (constructive interference). In contrast, the excited scattering modes of the meta-atoms collectively create destructive interference towards the free space within the detection numerical aperture (NA), resulting in an imaginary Poynting vector (Fig. 1e). This destructive interference, termed dark mode of the metasurface, is represented by $\left|\sum_{(i,j)}\mathbf{E}_{ij}\right|^2$ term in Eq. (2) in the absence of a nanoscatterer. When a nanoscatterer approaches, it disturbs the local resonance mode of the meta-atom elements and alters the phase profile of the sub-diffraction grating (Fig. 1f). The design of the meta-atom is chosen at a resonance mode, such that the amplitude $A_{ij}$ and phase $\phi_{ij}$ of the resonance mode are sensitive to the nearby environmental change due to the presence of the nanoscatterer, and the perturbed mode within the sub-diffraction array radiates towards the detection optics. Considering the perturbed electric fields of meta-atom elements, Eq. (2) is modified as

$$\left|\sum_{(i,j)}\mathbf{E}_{ij}{'}\right|^2 + 2\sum_{(i,j)}\left|\mathbf{E}_{ij}{'}\right|\left|\mathbf{E}_s\right|\cos(\alpha_{ij}) + \left|\mathbf{E}_s\right|^2 \tag{3}$$

The bright mode of the metasurface $\left|\sum_{(i,j)}\mathbf{E}_{ij}{'}\right|^2$ and $\left|\mathbf{E}_{ij}{'}\right|$ thereby generates an interferometric image with significantly enhanced contrast and SNR. Accordingly, the image contrast can be written as

$$C{\prime} = \frac{\left|\sum_{(i,j)}\mathbf{E}_{ij}{'} + \mathbf{E}_s\right|^2 - \left|\sum_{(i,j)}\mathbf{E}_{ij}\right|^2}{\left|\sum_{(i,j)}\mathbf{E}_{ij}\right|^2} \tag{4}$$

We implement such a plasmonic metasurface with sub-diffraction hexagonal meta-atom arrays, where degenerate plasmonic hybridized modes achieve a strong scattering condition (see details in Supplementary Section 1).

### Metasurface characterization

The plasmonic hexagonal meta-atom arrays have dimensions of $h = 60$ nm in height and $d = 70$ nm in diameter. As shown in Fig. 2a, meta-atom arrays are embedded in the polymethyl methacrylate (PMMA) on the silica substrate. The incident direction is set to be just one direction from the top medium (air) with azimuthal angle $\varphi = 180°$ and altitude angle $\theta_{in} = 75°$ (see details in Supplementary Section 2,3). Figure 2b presents the absorption cross-section ($\sigma_{ab}$) spectra of a meta-atom element in an array with different periodicities $\Lambda$ under p-polarization. The eigenmode analysis results are depicted by gray lines. In the visible wavelength regime, a meta-atom has mainly three transverse plasmon

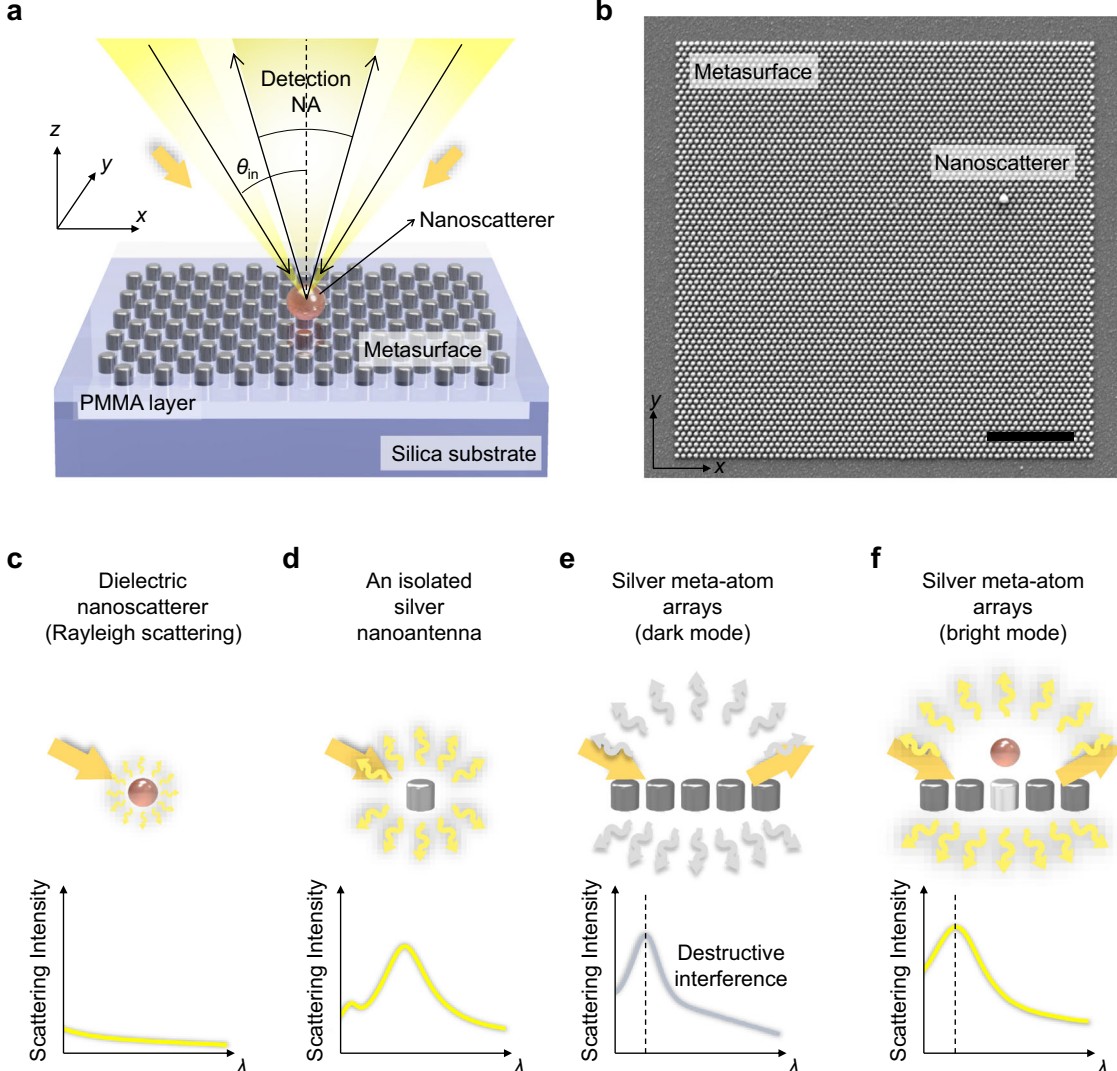

**Fig. 1 | Working principle of the MAD-iSCAT system. a** A schematic of MAD-iSCAT imaging (PMMA: polymethyl methacrylate and NA: numerical aperture) when the illumination angle is $\theta_{in}$. The arrows for the reflection lights are omitted. **b** Scanning electron microscopy image of metasurface without PMMA layer. Scale bar, 2 μm. **c**–**f** Light scatterings of (**c**) a dielectric nanoscatterer, (**d**) an isolated silver nanoantenna, (**e**) silver meta-atom arrays, and (**f**) silver meta-atom arrays with a detection target. The straight yellow arrows represent incident and reflection lights. In (**e**, **f**), the reflection is formed by the scattered lights from meta-atom arrays. The curved yellow arrows indicate the scattered lights.

(TP1, TP2, and TP3) modes and one longitudinal plasmon (LP1) mode[43]. The charge distribution patterns for each plasmon mode are presented in Fig. 2c. When $\Lambda$ reduces, wavelength shifts of longitudinal modes and transverse modes behave differently along their coupling direction of the plasmonic hybridization[44]. The theoretical analysis confirms the existence of degenerate plasmon modes arising from the antibonding longitudinal modes and bonding transverse modes at the sub-diffraction $\Lambda$. Here, we refer to such degenerate plasmon modes, which result in higher $\sigma_{ab}$ and $\sigma_{sc}$ as super-scattering plasmon (SSP) modes, as shown in Fig. 2b,d[45]. In contrast, for an isolated silver nanoantenna of the same dimension, TP2 and LP1 modes are spectrally separated (see details in Supplementary Section 4).

We design $\Lambda$ to be 145 nm such that the meta-atoms achieve the degenerate mode of TP2 and LP1 modes. The $\sigma_{sc}$ of SSP mode with $\Lambda = 145$ nm and $\lambda = 450$ nm is calculated to be $2.3 \times 10^{-14}$ m$^2$, surpassing those of non-degenerate dipolar and quadrupole modes which are typically broadened and weakened in sub-diffraction arrays compared to the isolated case[46]. This implies that the dense sub-diffraction meta-atom array can exhibit a strong scattering reference signal to form an interferometric image when SSP mode is

coupled into free space by the detection target. However, without a nanoparticle to be detected, the deep subwavelength $\Lambda$ ensures minimal background scattering under dark-field illumination. In addition, a shorter periodicity with a larger fill factor is preferred for a higher detection yield of the nanoparticles. It is also noteworthy that SSP mode is useful for particle detection since we can induce denser charge distribution on the interfacing surface where the detection target will be placed.

Since in real experiments, we use unpolarized light with a dark-field illumination with $0° \leq \varphi \leq 360°$, we also study $\sigma_{ab}$ and $\sigma_{sc}$ properties with different incident directions $\varphi$ and polarizations, as detailed in Supplementary Section 5. Under the same p-polarization, the spectral characteristics of the meta-atoms do not change significantly along the incident direction. In addition, when $\Lambda = 145$ nm and $\lambda = 450$ nm, $\sigma_{sc}$ with p-polarization dominates over that with s-polarization which is $3.5 \times 10^{-15}$ m$^2$. These findings support that our metasurface can be applied to a simple optical setup, such as a dark-field objective lens, without the need for precise polarization and illumination direction controls, facilitating easy use. Figure 2e shows absorbance peaks of 84% and 38% for p- and s-polarizations at

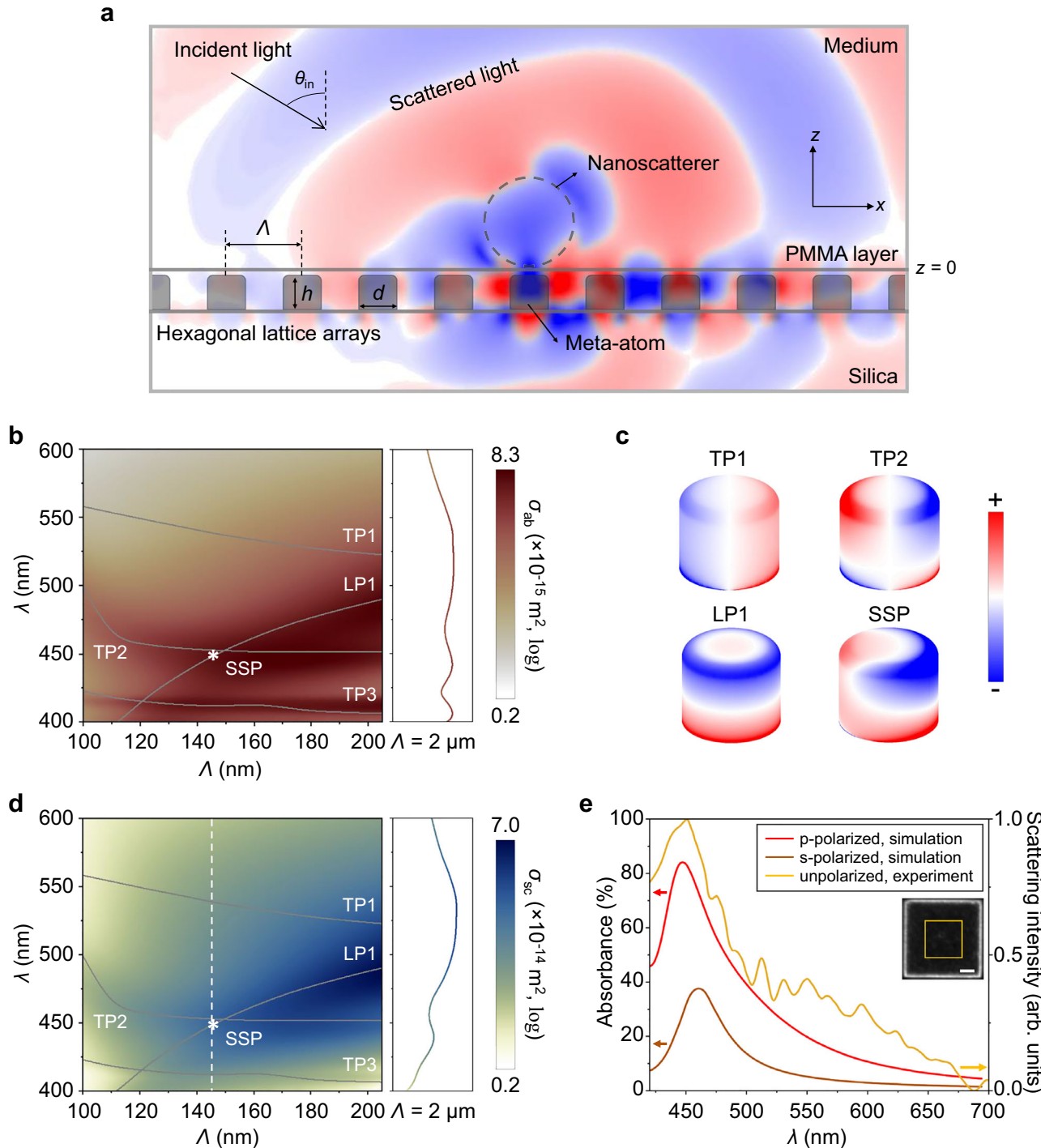

**Fig. 2 | Metasurface characterization. a** Design of the hexagonal meta-atom arrays for MAD-iSCAT imaging. The silver meta-atom arrays are placed on the silica substrate with an additional PMMA protection layer. The side view at $y = 0$ shows the geometrical parameters of meta-atom arrays: height ($h$), diameter ($d$), and periodicity ($\Lambda$). The incident angle $\theta_{in}$ corresponds to the dark-field illumination angle used for MAD-iSCAT. The exemplary scattered electric field distribution is obtained with a polystyrene nanoscatterer whose diameter is 200 nm, illuminated by a p-polarized light. In the presence of a nanoscatterer, the amplitude and the phase of meta-atom elements are modified by $\Delta A$ and $\Delta \phi$, respectively. **b** Absorption cross-section ($\sigma_{ab}$) spectra of silver meta-atom elements with different $\Lambda$ when $d = 70$ nm, $h = 60$ nm, $\theta_{in} = 75°$, and $\varphi_{in} = 180°$ under p-polarization. The transverse plasmon (TP) mode and longitudinal plasmon (LP) mode of meta-atom elements are depicted by gray lines. MAD-iSCAT uses the super-scattering plasmon (SSP)

mode when $\Lambda = 145$ and $\lambda = 450$ nm, which is denoted by the white star mark. **c** Surface charge distribution for each plasmon eigenmode and SSP mode of silver meta-atom elements. **d** Scattering cross-section ($\sigma_{sc}$) spectra of silver meta-atom elements with different $\Lambda$ when $d = 70$ nm, $h = 60$ nm, $\theta_{in} = 75°$, and $\varphi_{in} = 180°$ under p-polarization. In (**b**, **d**), $\sigma_{ab}$ and $\sigma_{sc}$ with $\Lambda = 2$ μm are also shown for comparison with those from an isolated silver nanoantenna (see details in Supplementary Section 4). **e** Absorbance spectra of meta-atom arrays are obtained by a numerical simulation using different incident polarizations. A scattering spectrum of meta-atom arrays is experimentally measured within the area depicted in the inset, for comparison with a $\sigma_{sc}$ spectrum in (**d**) denoted by the white dashed line. The inset presents a dark-field image of the meta-atom arrays with a broadband halogen lamp. Scale bar, 2 μm.

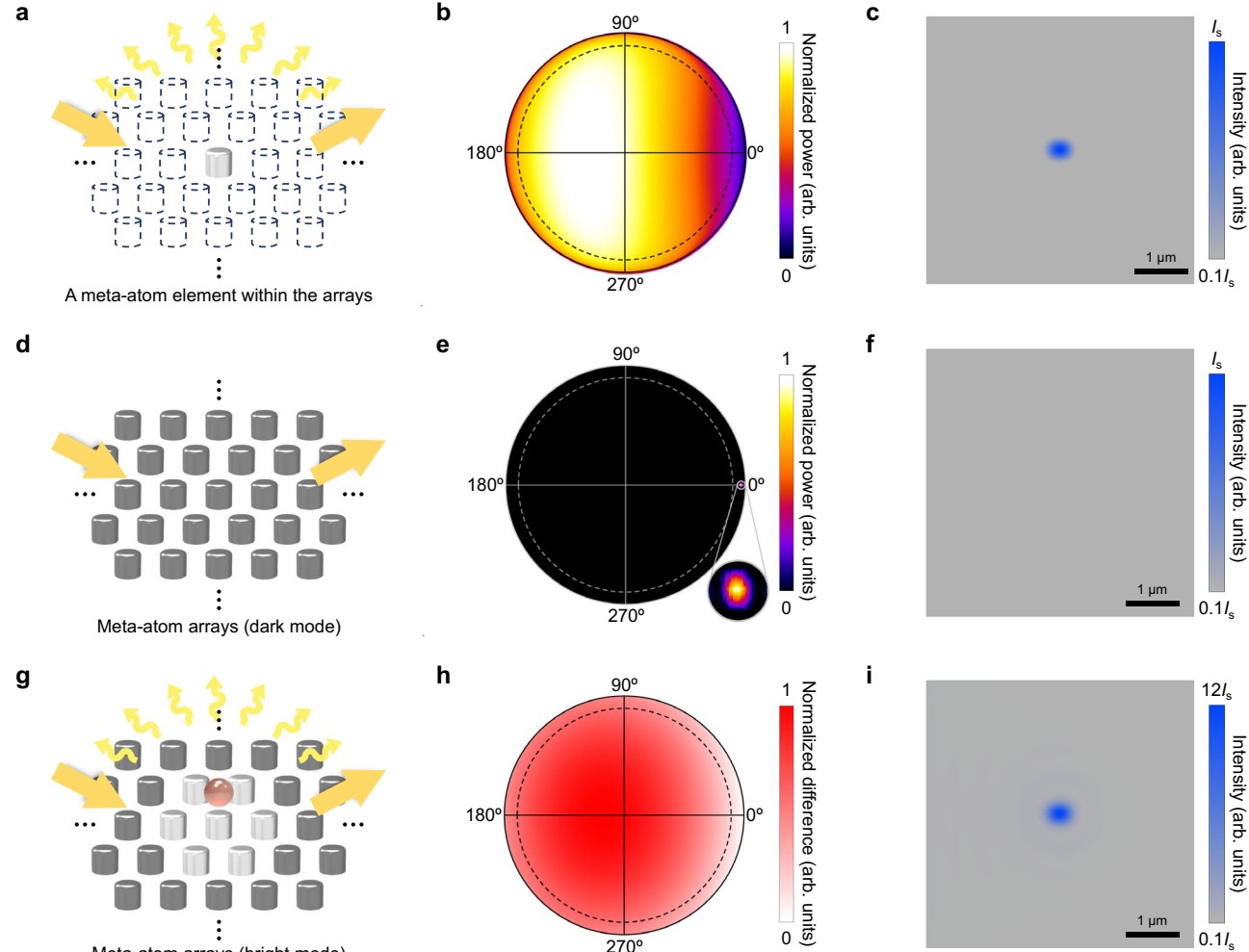

**Fig. 3 | Enhancing the scattering signal of a particle by sub-diffraction silver meta-atom arrays. a** An effective scattering of a single meta-atom element within the arrays with $\theta_{in} = 75°$. **b** A polar plot depicting the far-field scattering pattern of the single meta-atom element at $\lambda = 450$ nm in the upper half-space. The dashed line represents the angle $\theta$ of the detection NA. **c** A simulated image of a single meta-atom element at $\lambda = 450$ nm using the radiation pattern in (**b**). **d** A dark-field image of the effective single meta-atom element using NA = 0.9, which corresponds to the collection angle $\theta = 64°$. Under dark-field illuminations, each meta-atom element within the array exhibits an identical radiation power (see Fig. 3b). However, they vary in oscillation phases, determined by their horizontal positions on the $xy$-plane with respect to the incoming light along the $x$-axis. The total radiation

detection target. The inset represents a zoomed-in image of the reflected beam in the far-field. **f** A simulated image of a dark mode of meta-atom arrays using the radiation pattern in (**e**). **g** A bright mode of meta-atom arrays with a detection target. **h** A polar plot of the differential far-field radiation pattern in the upper half-space between with and without a polystyrene (PS) nanosphere whose diameter is 200 nm. **i** A simulated scattering image when a PS nanosphere is placed on the meta-atom arrays as shown in (**g**). The color bar is linearly scaled to a maximum of $12I_s$.

$\lambda = 450$ nm, respectively. Those absorbance peaks from numerical calculations match the peak wavelength of the scattering background measurement from our metasurface.

## The formation of a scattering image by a metasurface
The bright mode of the metasurface determines the total detectable intensity in a particle-sensing event. To understand the radiation mode of a meta-atom element inside the array, we obtain its far-field radiation pattern in the upper half-space by using electric polarization components extracted from the numerical simulation when $\varphi = 180°$ and $\theta_{in} = 75°$, as shown in Fig. 3a, b[47]. Note that the scattering radiation properties of the hybridized meta-atom in the sub-diffraction arrays would be different from a non-hybridized silver nanoantenna and cannot be directly measured in the experiment. Figure 3c shows a dark-field image of the effective single meta-atom element using NA = 0.9, which corresponds to the collection angle $\theta = 64°$. Under dark-field illuminations, each meta-atom element within the array exhibits an identical radiation power (see Fig. 3b). However, they vary in oscillation phases, determined by their horizontal positions on the $xy$-plane with respect to the incoming light along the $x$-axis. The total radiation

pattern from such meta-atom arrays in Fig. 3d is obtained using the phased array antenna theory. The constructive interference from the meta-atom arrays (19.4 × 19.3 μm²) forms a beam along the reflection of dark-field illumination ($\varphi = 0°$ and $\theta = 75°$) as shown in Fig. 3e. It should be emphasized that the element factor of the phased arrays determines the radiation properties of each meta-atom, while the array factor governs the constructive or destructive interference, thereby determining the direction of the Poynting vector into free space. Figure 3f shows that the collective scatterings from phased array meta-atoms do not exhibit energy propagation into the detection NA, leading to a significant reduction in the background signal, in contrast to Fig. 3c. For practical implementation with finite-sized metasurfaces, we also examine the effect of the array size on the background suppression with different numbers of meta-atom elements, as detailed in Supplementary Section 6.

As shown in Fig. 3g, when single or multiple meta-atom elements interact with the detection object, the dark mode can be turned into the radiation mode locally due to the introduced amplitude deviations $\Delta A$ and phase retardations $\Delta\phi$. These perturbations induce discontinuity in phased arrays, leading to radiation within the detection

NA. Figure 3h depicts the far-field radiation difference between such local radiation induced by a 200-nm polystyrene (PS) nanosphere and the dark mode. The local radiation results from the interference of EM fields from meta-atom elements and the scatterer (see details in Supplementary Section 7). Figure 3i shows an interference image of a 200-nm PS nanosphere on top of the metasurface. For better visualization, the color bars of Fig. 3f, i are linearly scaled to those of Fig. 3c. Here, the scatterer not only adds its own scattering to the total radiation pattern but also modulates the radiation power and phase of neighboring meta-atom elements. Supplementary Section 8 shows $\Delta A$ and $\Delta\phi$ introduced by the nanosphere as a function of the sizes. The phase perturbation of a meta-atom near the nanosphere is obtained by performing the numerical simulation of the metasurface with and without the particle and calculating the electric field phase difference (see methods for numerical calculation details).

Consequently, the significant difference in amplitude and phase of the meta-atoms between the bright mode and the dark mode leads to a greatly improved image contrast compared to the case where a dielectric nanosphere is situated within a homogeneous medium or a planar interface made by conventional media[39,40]. Supplementary Section 8 shows that the scattering intensity in the image as a function of particle diameter scales much more slowly than pure Rayleigh scattering. This is because the dominant scattering intensity arises from the interference between the meta-atom arrays, the background, and the detection particle rather than from the detection particle itself (see details in Supplementary Section 9)[39,40].

### Experimental results

To experimentally demonstrate the scattering intensity and contrast enhancement of MAD-iSCAT, we fabricate hexagonal silver meta-atom arrays by nanoimprint lithography on a silica substrate with an additional PMMA protection layer and use PS nanobeads as scatterers. We utilize a broadband supercontinuum laser as the light source to produce interference images of PS nanobeads at different wavelengths ranging from 430 to 700 nm with a dark-field objective in an air environment (see details in Supplementary Section 10). The PS nanobeads have varied sizes of 200, 100, 62, and 45 nm. Figure 4a–d represents the enhanced scattering intensities of PS nanobeads on meta-atom arrays. Figure 4e, f compares those with the case of nanobeads on a bare PMMA layer for diameters of 200 and 100 nm, respectively. In Fig. 4g, h, we identify the same nanobeads with scanning electron microscopy (SEM) as the ground truth images (see details in Supplementary Section 11). In the case of 200- and 100-nm diameter nanobeads, the maximum enhancement factors ($I_{MS}/I_{PMMA}$) are found to be $11.44 \pm 2.08$ and $31.45 \pm 5.41$ at $\lambda = 580$ and 490 nm, respectively. Meanwhile, the enhancement factors at $\lambda = 450$ nm are $5.60 \pm 1.22$ and $20.83 \pm 4.50$, respectively. We experimentally confirm the scattering enhancement factors of 66.20 and 128.87 for nanobeads of 62- and 45-nm diameters at the resonant wavelength $\lambda = 450$ nm, respectively. The MAD-iSCAT metasurface exhibits significant scattering enhancement factors over visible wavelengths, as shown in Fig. 4i. Figure 4j presents the normalized detected intensity of the PS nanobeads on the meta-atom arrays and PMMA as a function of nanobead size, with the cubic power of particle size shown for comparison (see details in Supplementary Section 12). We also conduct MAD-iSCAT imaging of small nanoparticles in aqueous environments, as shown in Fig. 5a. The imaging system is constructed with a custom axicon-based water-immersion dark-field microscopy[48]. Fig. 5b, c compares MAD-iSCAT and regular iSCAT using PS nanobeads with diameters of 22, 41, and 81 nm. Regular iSCAT measurements are performed using a standard microscope and a laterally modulating piezoelectric nano-positioning stage with lock-in image detection. The iSCAT images are obtained by integrating 330 frames (effective exposure time (EET) of 116 ms). For those particle sizes, regular iSCAT yields image contrasts of $0.6 \pm 0.2\%$, $3.9 \pm 1.7\%$, and $23.9 \pm 3.1\%$ ($n = 127$,

137, and 38), respectively. In contrast, MAD-iSCAT demonstrates image contrasts of $14.9 \pm 4.5\%$, $165.6 \pm 32.7\%$, and $869.1 \pm 124.9\%$ ($n = 451$, 283, and 20) for the same particle sizes, employing a single-frame differential process (EET of $1\,ms \times 2 = 2\,ms$). MAD-iSCAT shows a contrast enhancement of $26 \pm 7.5$ for the 22 nm beads compared to regular iSCAT. This comparison highlights the improved sensitivity and high-throughput capability of MAD-iSCAT.

To demonstrate the sensitivity of MAD-iSCAT in biological nanoparticle detection scenarios, we image individual exosomes extracted from MDA-MB-231 human breast cancer cells. Figure 6a, b shows exemplary MAD-iSCAT images of exosomes in an aqueous environment. The size of each exosome is determined through mean-squared displacement analysis[4] of identified particles in subsequent frames (see details in Supplementary Section 13). Figure 6c shows the size-dependent image intensity of each detected exosome using MAD-iSCAT and Rayleigh scattering. The performance of MAD-iSCAT offers 1–2 orders of magnitude improvement for various sizes of exosomes. To further validate the sensitivity of MAD-iSCAT, we conduct experiments using ferritin, a protein with a molecular weight of ~440 kDa. With an exposure time of 1 ms (EET of 2 ms), Fig. 6d demonstrates the MAD-iSCAT measurement for ferritin molecules, with an observed contrast of $7.7 \pm 1.4\%$ ($n = 805$). Figure 6e presents an exemplary image showing individual ferritin molecules. The image SNR comparison shows that our approach demonstrates an SNR enhancement factor of 19.8 relative to conventional iSCAT (see details in Supplementary Section 14). This contrast and SNR enhancement for a protein illustrate the improved sensitivity of our method.

## Discussion

It is worth noting that previous plasmonic scattering microscopy primarily depended on field enhancement on metallic surfaces directly interacting with the detection targets[39]. Later, it was demonstrated that utilizing higher power and shorter wavelengths on bare glass substrates can lead to advancements in evanescent-wave-based sensing, broadening their applications[40,49,50]. However, the effect of surface roughness in iSCAT was not explicitly addressed in previous studies. MAD-iSCAT underscores that the enhanced image contrast arises not only from the field enhancement effects near the silver surfaces but also from the additional interference with the radiation modes of silver meta-atom arrays. The perturbed electric field from meta-atoms generates strong interferometric signals, which exhibit an approximate cubic power law dependence that persists even for comparably larger particle sizes where Rayleigh scattering dominates[39,40,50]. The laser power employed is sufficiently low to avoid significant photothermal effects and our protective layer also minimizes the heating effect on detection targets and prevents photodamage to silver meta-atoms[51].

In the current MAD-iSCAT implementation, due to fabrication errors, the meta-atoms exhibit deformation from their original designs. This leads to a higher dark mode background level as well as a degraded TP2 mode in silver meta-atoms, which is reflected in the broader wavelength response shown in Fig. 2e compared to the ideal meta-atom resonance. Nevertheless, the noise level of MAD-iSCAT does not significantly compromise the enhanced scattering signals, effectively demonstrating the amplification of scattering signals from small nanoparticles. The noise analysis of our system is detailed in Supplementary Section 15.

In summary, we introduce an interferometric imaging method, MAD-iSCAT, which employs a metasurface comprising plasmonic meta-atoms to significantly enhance the small nanoparticle detection sensitivity. Unlike diffractive plasmonic pixels, our approach harnesses sub-diffraction arrays to produce amplified interference signals, inspired by phased array antenna theory[52]. We introduce a plasmon hybridization within the sub-diffraction silver meta-atom arrays, theoretically investigate the generation of interferometric images, and provide experimental validations of an enhanced contrast and SNR. As a result, the

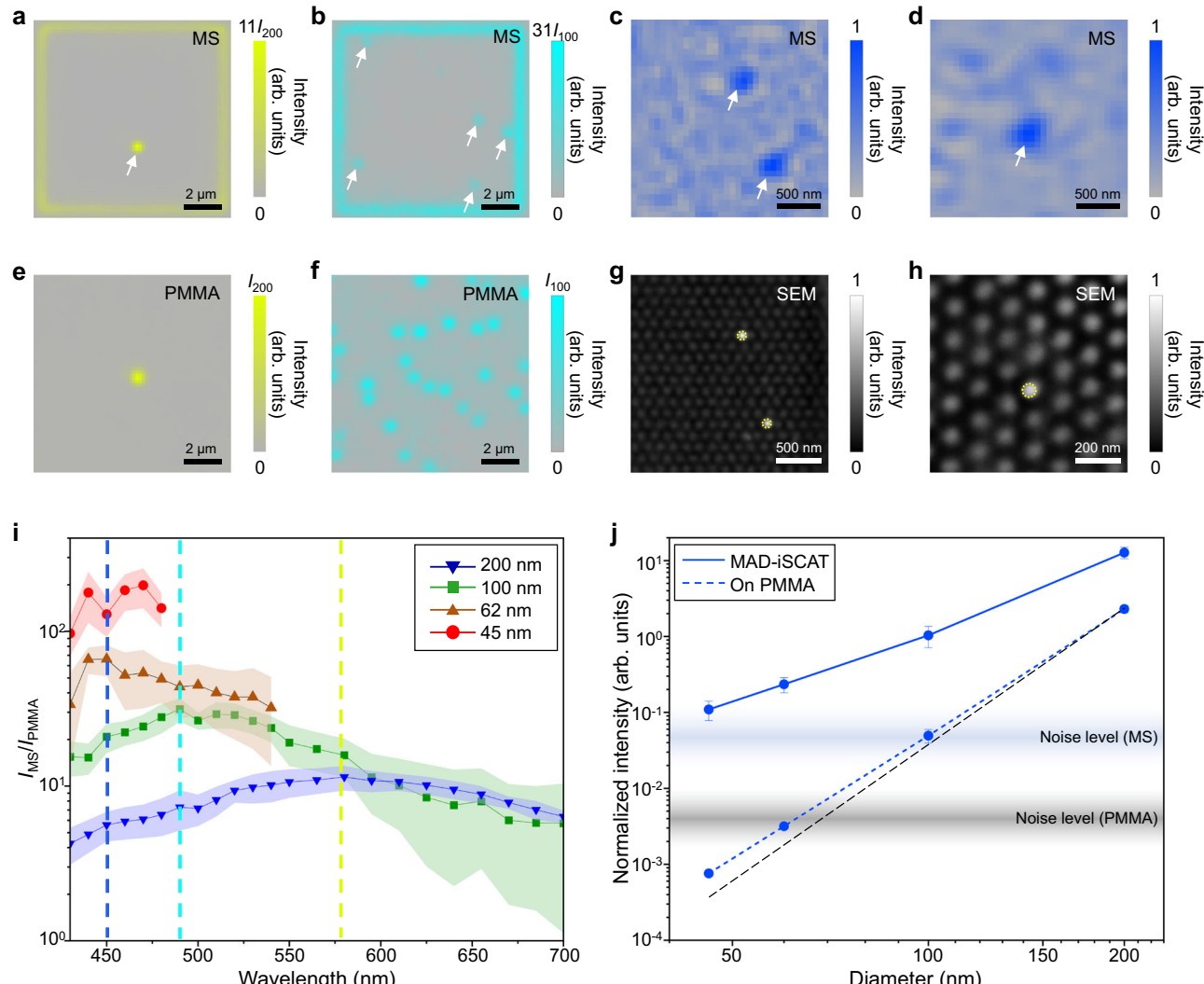

**Fig. 4 | Amplification of interferometric scatterings from dielectric nano-particles using the metasurface. a–d** Scattering images of polystyrene (PS) nanobeads on the surface of the metasurface (MS, 9.57 × 9.54 μm²) with varying sizes: (**a**) 200, (**b**) 100, (**c**) 62, and (**d**) 45 nm in diameter. **e, f** Scattering images of PS nanobeads on the PMMA layer with sizes of (**e**) 200 and (**f**) 100 nm in diameter. The color bars in (**a**) and (**b**) are linearly scaled by using the maximum detected intensity ($I_{200}$ and $I_{100}$ for 200- and 100-nm nanobeads, respectively) on bare PMMA for each case. Wavelengths of 580, 490, and 450 nm are used for (**a, e**), (**b, f**), and (**c, d**), respectively. **g, h** SEM images of the same PS nanobeads on the surface of the

metasurface in (**c**) and (**d**), respectively. **i** Enhancement factors comparing $I_{MS}$ with $I_{PMMA}$ for different nanobead sizes at visible wavelengths. The $I_{MS}$ with diameters of 62 and 45 nm are compared with simulated values of $I_{PMMA}$. **j** Normalized scattering intensities dependent on the nanobead's diameter. The wavelength is at $\lambda$ = 450 nm. Mean values and standard deviations are obtained for varied sizes ($N_{200}$ = 12, $N_{100}$ = 10, $N_{62}$ = 9, and $N_{45}$ = 5). The nanoparticles are identified with ground truth data of fluorescence and SEM images. The gray dashed line represents the sixth power dependence.

interferometric image contrast of dielectric nanoparticles, exosomes, and proteins is remarkably improved. It is important to note that the current MAD-iSCAT has a detection volume more constrained than conventional iSCAT and suffers from fabrication imperfections, array inhomogeneities, and finite-size effects. Nonetheless, MAD-iSCAT combines the advantages of iSCAT and meta-atom arrays, opening the opportunities for optimization in label-free sensing applications. MAD-iSCAT has the potential to dramatically improve single-molecule mass imaging throughput by enhanced fabrication approaches, such as self-assembly[53] and large array geometries for expanded field-of-view and suppressed background noise. The method also holds promise for label-free interferometric super-resolution imaging[41,54].

## Methods
### Metasurface fabrication
A plasmonic metasurface that consists of silver meta-atom arrays on a silica substrate is fabricated based on nanoimprint lithography (NIL),

reactive ion etching (RIE), and metal e-beam evaporation. A mother mold of the designed meta-atom arrays for NIL is fabricated on a silicon wafer using standard e-beam lithography. The mother mold contains 360 copies of the meta-atom arrays to ensure a sufficiently large field-of-view during imaging. The silica substrate is prepared with two spin-coated resist layers. One is a PMMA underlayer as the liftoff layer and the other is a UV-curable resist layer at the top. NIL is performed to define the meta-atom structures on the UV resist layer. After curing the UV resist and releasing the mother mold, the residual UV resist and the underlying PMMA underlayer are both etched by RIE to expose the bottom silica substrate. 60 nm silver is evaporated onto the sample afterward with e-beam evaporation, followed by a lift-off process. After NIL, a PMMA reflow step is employed to planarize the metasurface as well as to protect the silver meta-atoms from oxidation and potential damage. A thin PMMA spin-coating process is repeated 3 times on the metasurface to produce a conformal coating that fully encloses the meta-atoms while maintaining a flat top surface. The total thickness of

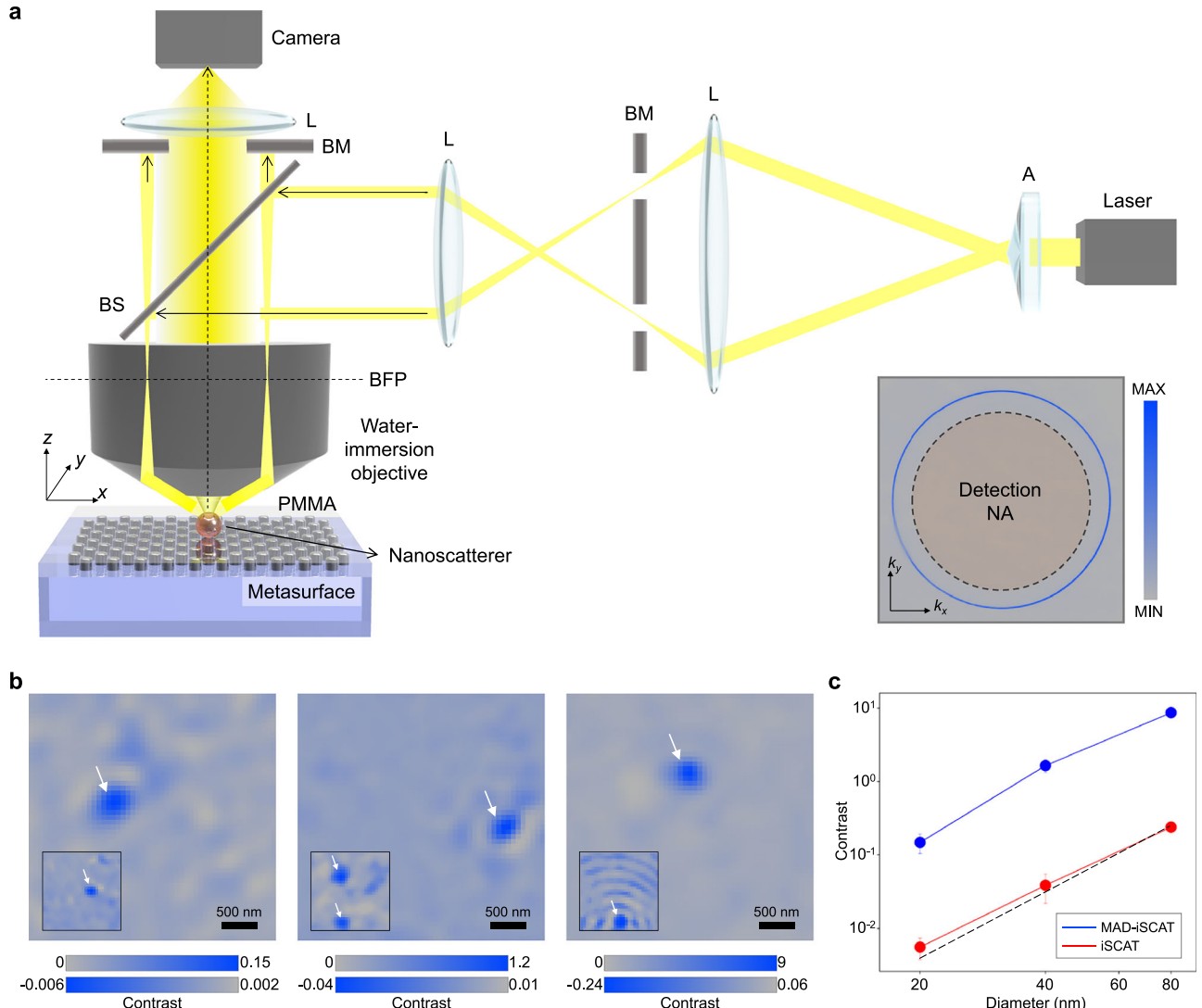

**Fig. 5 | MAD-iSCAT imaging in aqueous environments. a** A schematic of the optical setup for the MAD-iSCAT in aqueous environments (A: axicon, L: lens, BM: beam block, BS: beam splitter, and BFP: back-focal plane). The right inset represents the measured BFP image of the incident beam and the detection NA. Solid arrows along the yellow beam path indicate the incident and reflected beams, while the dashed arrow represents the scattered light. **b** MAD-iSCAT (top color bars) and iSCAT (insets, bottom color bars) images of PS nanobeads having the diameters of 22, 41, and 81 nm in an aqueous environment. **c** The comparison between the image contrasts of PS nanobeads detected by MAD-iSCAT (blue) and iSCAT (red). The black dashed line represents the cubic power. White arrows indicate the nanobeads on the surface.

the PMMA planarization layer is around 75 nm. The 60 nm meta-atoms remain close to the substrate surface.

## Experimental setup

A supercontinuum laser light source (SuperK Extreme EXB-6, NKT Photonics) is utilized to acquire MAD-iSCAT images at wavelengths ranging from 430 to 700 nm. The laser power of the dark-field ring at the center wavelength of $\lambda = 450$ nm is measured to be 0.15 mW. The light source is delivered to the metasurface surface through a dark-field objective lens (UMPlanFl, ×100, 0.9 NA, Olympus). The incident angle of the dark-field objective lens is 75°. An electron-multiplying charge-coupled-device camera (iXon$^{EM}$+ DU-897, Andor Technology) with a 100-ms exposure time and 160 accumulations is employed to obtain a large dynamic range for recording background noise and strong scattering signals from meta-atom arrays. The scattering spectrum of the metasurface is measured using a spectrometer (Shamrock 303i, Andor Technology) and a halogen lamp as the illumination source. To verify the amplified scattering signals from small nanoparticles, PS nanobeads with diameters of 200 (Nanosphere™ Size

Standards, 3200 A, Thermo Fisher Scientific), 100 (FluoSpheres™ carboxylate, F8803, Invitrogen), 81 (Nanosphere™ Size Standards, 3080 A, Thermo Fisher Scientific), 62 (Nanosphere™ Size Standards, 3060 A, Thermo Fisher Scientific), 45 nm (FluoSpheres™ carboxylate, F8792, Invitrogen), 41 (Nanosphere™ Size Standards, 3040 A, Thermo Fisher Scientific), and 22 (Nanosphere™ Size Standards, 3020 A, Thermo Fisher Scientific) are utilized after dilution. The size deviation for each size is provided as 6, 7, 3, 3, 8, 4, and 2 nm by the manufacturer, respectively. Each PS bead solution is drop-casted onto the metasurface after cleaning the interfacing surface with deionized water. Image processing is conducted using home-built MATLAB R2023b codes and ImageJ functions for data analysis and presentation. In the case of 62-nm nanobeads, applying background subtraction results in a scattering image with enhanced intensity without requiring temporal differentiation (Fig. 4c). In the case of 45-nm nanobeads, we apply temporal differentiation (Fig. 4d). In Fig. 4j, each scattering intensity is normalized by that of an isolated silver nanoantenna. For the exosome detection, we use the water-immersion objective lens (UPlanSApo, ×60, 1.2 NA, Olympus). The laser power of the dark-field

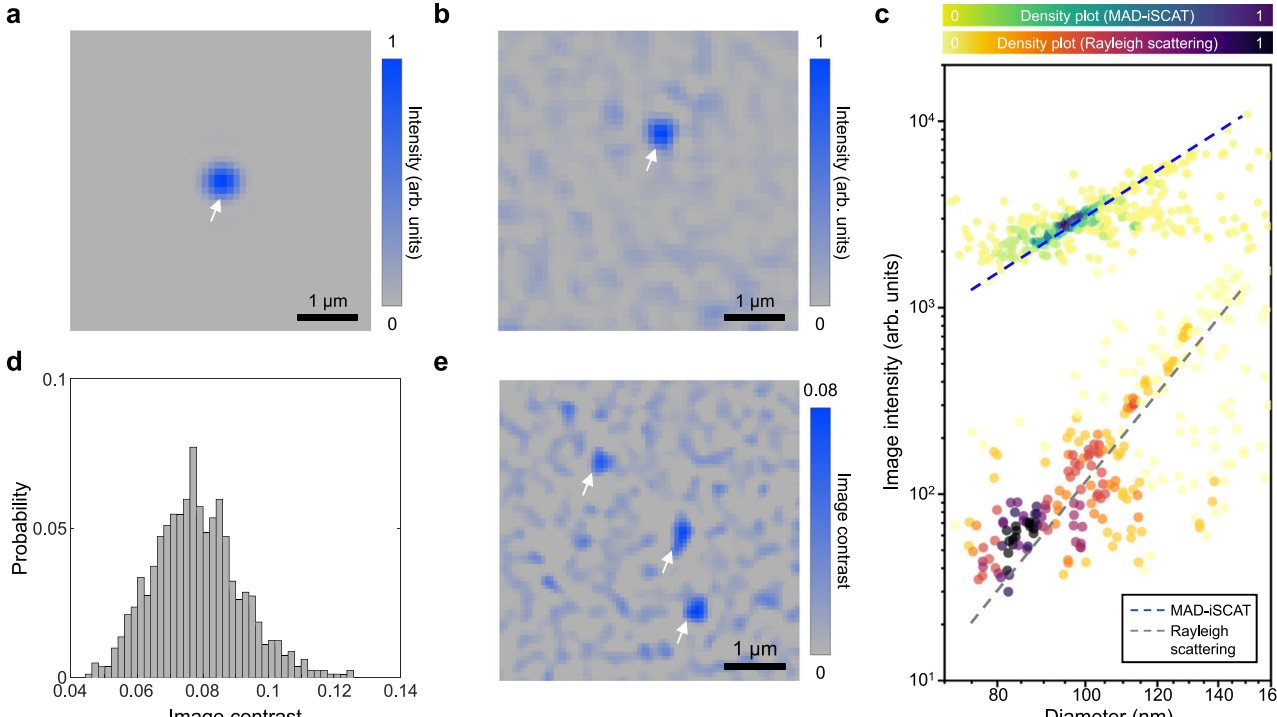

**Fig. 6 | MAD-iSCAT imaging of exosomes and ferritin molecules. a, b** MAD-iSCAT images of exosomes obtained by (**a**) the simulation and (**b**) the experiment with a diameter of 80 nm. **c** Image intensity of exosomes dependent on different sizes. Density scatter plots for the case of MAD-iSCAT represent the image intensity of exosomes and their size using 397 diffusion trajectories. Density scatter plots for the case of Rayleigh scattering represent the image intensity of exosomes and their size using 303 diffusion trajectories. Blue and gray dashed lines in (**c**) represent the theoretical scattering intensities of exosomes that can be obtained on the metasurface (MAD-iSCAT) and in the aqueous solution (Rayleigh scattering). **d** Histogram of contrast obtained from ferritin using MAD-iSCAT. **e** An exemplary MAD-iSCAT image of ferritin molecules. White arrows indicate the detection objects on the surface.

ring at $\lambda$ = 450 nm is measured to be 1.1 mW. An sCMOS camera (ORCA-Flash4.0, C13440-20CU, Hamamatsu) with an exposure time of 487.218 μs is employed. The dark-field ring at the back focal plane of the water-immersion objective is generated and its size adjusted using an axicon and two lenses in series, as illustrated in Fig. 5a. An example of the dark-field ring is shown in the inset of Fig. 5a. For MAD-iSCAT measurements for PS nanobeads and ferritin proteins (F6754-1VL, Sigma-Aldrich) in aqueous environment, we employ an identical optical setup as used for exosome detection with an additional ×1.6 magnifier and an exposure time of 1 ms. The PMMA surface is incubated with poly(allylamine hydrochloride) (PAH) to facilitate adhesion. No frame averaging is applied. Images are processed using stabilization, Gaussian filtering to reduce noise, and background correction. For the 80-nm PS nanobeads, the contrast is obtained using frames separated by 10 frames, whereas for the 20- and 40-nm beads, neighboring frames are used. Incident angles are selected to satisfy the same plasmon modes excitation with $\mathbf{k}_x = \mathbf{k}_0\sin75° = \mathbf{k}_{water}\sin47° = \mathbf{k}_{PMMA}\sin40°$.

For iSCAT measurement, we use a studied iSCAT setup incorporating a nano-positioning stage (Nano-Drive 85, Mad City Labs) to enable translational modulation of scatterers on a glass substrate[7]. This setup allows us to isolate the scattered signal from the background. An sCMOS camera (ORCA-Flash4.0, C13440-20CU, Hamamatsu) is employed. The experiments are conducted with an exposure time of 351 μs and stage modulation at 25 Hz. The five periods of translation modulation are utilized to acquire the scatter signal by eliminating the background. This is achieved by discarding the frames captured during stage translation, resulting in an integration of 330 frames. A high-NA objective lens (UAPON, 100×, 1.49 NA, Olympus) is used with an additional ×1.6 magnifier and the glass surface is treated with poly-D-lysine in PBS (1 mg/mL) to facilitate adhesion of negatively charged PS beads.

## Sample preparation

Human breast cancer cell line MDA-MB-231 (HTB-26) is obtained from the American Type Culture Collection and cultured in Dulbecco's Modified Eagle Medium (DMEM; Gibco) supplemented with 10% fetal bovine serum (Sigma-Aldrich). Small extracellular vesicles enriched in exosomes are purified from the conditioned medium (CM) using differential centrifugation. Briefly, CM is collected from cells incubated in serum-free DMEM for 24 h, precleared by centrifugation at 500 × g for 15 minutes, and then at 10,000 × g for 20 min. Extracellular vesicles are passed through a 0.22-μm filter, pelleted by ultracentrifugation at 110,000 × g for 120 min, and washed in PBS using the same ultracentrifugation conditions. The exosomes suspended in phosphate-buffered saline (PBS) solution are sealed in an aqueous chamber made of the MAD-iSCAT metasurface and a coverslip so that their diffusion dynamics can be captured.

## Simulations

Finite-element analysis of EM fields for the metasurface is conducted using COMSOL 6.0 Multiphysics. Initially, a unit cell ($\Lambda \times \sqrt{3}\Lambda$ nm²) with a height of 1 μm is employed, applying periodic boundary conditions laterally and perfectly matched layer (PML) boundary conditions along the vertical direction. The near-field distributions are shown in Supplementary Section 16. The analysis aims to investigate $\sigma_{ab}$ and $\sigma_{sc}$ spectra of meta-atom elements, the absorbance spectrum of meta-atom arrays, charge distribution, and eigenmode analysis. Field intensity normalization is performed using reference calculations conducted in the absence of meta-atom arrays. Incident light with different polarizations, directed from the top medium (air) at $\varphi = 180°$ and $\theta_{in} = 75°$, is considered. The mesh size ranges from 1 to 25 nm. Refractive indices of silver, PS, PMMA, and SiO₂ are taken from the references[55–58]. In Fig. 2b, d, the wavelength and periodicity are swept

with different step sizes to reduce the simulation time and interpolation is applied afterward. Subsequently, to examine the formation of interferometric scattering images by the metasurface, a larger array of unit cells, laterally spanning $10\Lambda$ along the $x$-axis and $5\sqrt{3}\Lambda$ along the $y$-axis is employed. The EM field distribution of meta-atom arrays without a detection nanosphere is calculated using Floquet periodic boundary conditions for in-plane directions. Scattered EM field distribution by a detection target is obtained by replacing Floquet periodic boundary conditions with PML boundary conditions[21]. Here, the phase perturbation of the nanoantenna caused by the particle was obtained by the phase difference of the electric component in the meta-atom between the first and second FEM calculations. The mesh size ranges from 1 to 50 nm. In numerical simulation, the edges of the nanoantenna are truncated to be round-shaped, considering practical experimental constraints. Electric polarization components extracted from the finite-element analysis are utilized to generate far-field properties of the detection target and neighboring meta-atom elements[45]. Here, far-field properties of the remaining meta-atom elements are confirmed to be almost identical as a result of the first simulation with the Floquet boundary condition. The far-field radiation pattern is calculated at a distance of 2 mm from the array center. For sufficiently large meta-atom arrays, edge scattering nearly interferes with light signals within the array. A home-built MATLAB code is utilized to generate a collective scattering of meta-atom arrays by adding more meta-atom elements for larger array sizes, whose far-field properties remain unmodified by the detection target.

## Data availability

The representative datasets and code used to generate the results of this study are accessible through the Figshare repository (https://doi.org/10.6084/m9.figshare.30812684). Due to file size constraints, data supporting this work can be obtained from the corresponding author upon request.

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

## Acknowledgements

This work was supported by the National Science Foundation (Grant no. CBET-2348536). We extend our gratitude to Yu Xie (Department of NanoEngineering, University of California, San Diego, United States), Professor Andrea R. Tao (Material Science and Engineering Program and Department of NanoEngineering, University of California, San Diego, United States), Ayse Sahan (Department of Pharmacology, University of California, San Diego, United States), Professor Jin Zhang (Department of Pharmacology, University of California, San Diego, United States), and Shuai Feng (Materials Science and Engineering, School for Engineering of Matter Transport and Energy, Arizona State University, United States) for their valuable discussions and for providing samples for surface treatment.

## Author contributions

H.L. conducted the theoretical analysis of the metasurface. H.L. and J.Z. performed the experiment. P.H. and W.W. fabricated the metasurface. Z.N. and S.Y. provided the mother mold. Ming L., G.C., and A.L. helped during the experiment. S.B., L.C., and Mojie L. helped during the theoretical analysis. S.E.W. prepared the biological samples. Z.L. supervised the work. H.L., J.Z., Q.W., and Z.L. wrote the manuscript.

## Competing interests

The authors declare no competing interests.
