## [Transparent Peer Review file · Nature Communications]

Meta-amplified dark-field interferometric scattering microscopy

Corresponding Author: Professor Zhaowei Liu

Version 0:

Reviewer comments:

Reviewer #1

(Remarks to the Author)

The manuscript reports of an interferometric technique, MAD-iSCAT, to improve nanoparticle detection sensitivity. Conventional i-SCAT uses a glass slide to accommodate the sample and generate the reference beam, while in MAD-iSCAT this reference beam is generated using a plasmonic metasurface. By engineering the metasurface and the illumination, a dark-field detection scheme is demonstrated. The presence of a particle scatters the nearfield mode of the structure into the far-field. The authors report a modified contrast equation (Eqn. 5,6) because of an additional 'bright mode' arising from the meta-atoms in the presence of a scatterer. As a result, high-contrast images of nanoparticles with enhanced sensitivity over conventional iSCAT is quantified and demonstrated. The results are interesting and the proposed concept is similar to Nature Communications volume 13, 2298 (2022).

(1) The following is not clear from the manuscript with regards to image processing and the contrast equation arising from it: The detected signal in conventional iSCAT in the presence of a scatterer is $I_s = |E_r|^2 + |E_s|^2 + 2|E_s||E_r|\cos\alpha$, where $|E_r|$ is the reference, $|E_s|$ is the scattered and α is the phase difference between these fields. After subtraction of two successive images (with and without a particle) the equation for the detected signal is approximated as $2|E_s||E_r|\cos\alpha$, assuming that $|E_r|$ is constant between the different temporal frames. The contrast of the processed image is $C \approx (2|E_s||E_r|\cos\alpha)/I_{bg}$, where $I_{bg} = |E_r|^2$.

In MAD-iSCAT, because of the dark-field detection, ideally no light should reach the camera and hence, this subtraction scheme isn't possible, so $C \approx (|E_r|^2 + 2|E_s||E_r|\cos\alpha)/I_{bg}$. For high-visibility, the magnitudes of the interfering fields must be very close. But since for nanoparticles, $|E_s|$ is low, then $|E_r|$ must also be low if you are to get a good contrast image. So if you assume that $|E_r| \approx |E_s|$ and neglect $|E_r|^2$ or you subtract the background, then the processed image contrast $C \approx (2|E_s||E_r|\cos\alpha)/I_{bg}$, which is the contrast achievable using a conventional iSCAT. However, experimentally there will always be some light reaching the camera, also seen in Fig. 3f. But then again if you perform temporal subtraction, the contrast of the processed image will still be $C \approx (2|E_s||E_r|\cos\alpha)/I_{bg}$.

(a) Please elaborate as to how this Eqn. (5,6) in the manuscript is arrived at and

(b) also why is MAD-iSCAT different from Zhang et.al.s paper cited above? In that work also, light scattered from the coverslip's surface roughness interferes with the scattered light by the sample. It will be beneficial for the readers to clearly understand how you derived the contrast equation, what assumptions are made and how is it different from the above mentioned paper? This will bring the novelty of the paper evident.

(c) When will this assumption break? For e.g., the plot in Fig. 3l, when will they intersect? Of course, iSCAT is for the detection of nanoparticles. So this question is purely out of curiosity to see when these assumptions break.

(2) Lines 54-55: "However, as the light scattering cross-section scales with the sixth power of the particle's diameter..."

(a) Rayleigh scattering scales with sixth power of the particle radius and not diameter. Please check and correct.

(b) Fig. 4l – black dashed line represents cubic power. Is this plotted as a function of radius or diameter?

(c) Fig. 4j – gray dashed line represents sixth power dependence.

(d) Explain Fig. 4l and Fig. 4j based on Rayleigh scattering scaling inversely as sixth power of radius. Please mention the values of the scattering intensity at 50 nm, 100 nm, upto 200 nm for both MAD-iSCAT and Rayleigh scattering. It is provided in the plot, but it is easier to compute the dependence on particle size from numbers directly given.

- (3) Fig. 4k --- can you explain why the particle locations have negative pixel values? Isn't particle location supposed to be affecting the pixel value because of Gouy phase? Then why do they have positive values in the Mad-iSCAT?
- (4) Fig. 4f – it is mentioned that all particles are validated with ground truth fluorescence and SEM images.
 (a) Is this done for all the results shown in the manuscript?
 (b) How was it ensured that fluorescence signal didn't leak into the iSCAT measurements?
- (5) Since MAD-iSCAT is a near-field technique, it will be good to have a plot illustrating the field strength as a function of distance from the meta-atom surface.
- (6) It was not clear whether the SSP mode was utilized for all the experimental results demonstrated in the manuscript.
- (7) Fig. 2a, can you provide the simulation results with and without the nano scatterer. This will aid the reader to see how the scatterer affects the confined mode.
- (8) Fig.3 - What is the spatial extent covered by the polar plots with respect to the meta-atom array? Maybe add a scale bar to highlight it and also, where exactly on the meta-atom array is this measurement carried out? Is it at the intersection of the dotted lines in Fig. 3a?
- (9) Fig.1(c-f) – a suggestion is that the plots can be better represented. For example, wavelength range along the x-axis will be useful as these meta-atoms do not operate at all wavelengths. Another point of confusion was the grey plot in the case of the dark-mode. This can be understood after reading the manuscript, but a bit more detailed explanation in the figure caption will also help.
- (10) Though the array spans a few microns, what is the field-of-view achievable (FoV)? Is it $9.6 \times 9.3 \mu\text{m}^2$? Do you think you can enhance the FoV? It will be useful to highlight these issues in the manuscript.
- (11) MAD-iSCAT is a near-field detection technique. Conventional iSCAT can help isolate particles within the focal volume of the detection objective. Therefore, though it is obvious, highlighting the limitations of MAD-iSCAT is important especially for biologists.
- (12) Fig. 4j – please correct the legends in the plot.
- (13) Due to enhanced field strengths, can particles get trapped to the high field intensity?
- (14) Have you quantified any temperature changes due to plasmonic effects? Will this affect the biological samples under study?

Reviewer #2

(Remarks to the Author)

The manuscript entitled “Meta-amplified dark-field interferometric scattering microscopy” has undergone one round of peer review, and the authors have revised the manuscript in response to the reviewers' comments. After reviewing the revised version and the response letter, I find that the authors have addressed the earlier concerns in a satisfactory manner. A notable strength of this work lies in the central concept of exploiting the dark mode of the plasmonic meta-atom array. By engineering the metasurface so that its collective radiation is suppressed in the absence of objects, the authors achieve a background-free baseline. Upon the attachment of nanoparticles, local perturbations induce a transition from the dark mode to a bright radiation mode, resulting in a strong and detectable signal. This mechanism represents an original idea that clearly differentiates MAD-iSCAT from previously reported iSCAT approaches. The physical picture is compelling and provides a solid foundation for the reported contrast enhancements. I strongly recommend this work for publication after one further round of minor revisions.

1. On page 3, line 57, the statement ‘Interferometric imaging addresses this challenge by leveraging the interference of scattered light from nanoscale objects with reference light, yielding label-free images with high sensitivity and resolution’ seems to overstate the claim of ‘high resolution’. Interferometric imaging does not inherently surpass the diffraction limit; rather, it improves localization precision or effective resolution through enhanced signal-to-noise.
2. In Fig. 1a, the schematic illustrates a reflection geometry where the illumination incidence angle is drawn larger than the detection NA cone, suggesting complete separation between illumination and detection. However, for a 0.9 NA dark-field objective, the illumination annulus specified by the manufacturer is typically $\text{NA} = 0.8\text{--}0.9$, corresponding to $\sim 53^\circ\text{--}64^\circ$ in air, which partially overlaps with the collection cone ($0\text{--}64^\circ$). This discrepancy between the schematic, the stated 75° incidence angle, and the actual objective specification should be clarified. In principle, one could increase the laser input incidence angle beyond the nominal annulus by back focal plane engineering while restricting the detection NA to a smaller cone. If such a strategy was employed, it would be important for the authors to explicitly state this in the Methods, as it would resolve the apparent inconsistency.
3. Despite the conceptual novelty, one potential weakness of the approach is that achieving a perfect dark mode in practice

may be difficult. Fabrication imperfections, array inhomogeneities, and finite-size effects could lead to residual scattering from the meta-atoms themselves, limiting the reproducibility and robustness of the method. It would be helpful if the authors could briefly acknowledge this limitation in the manuscript, as it provides important context for the practical implementation of MAD-iSCAT. Although the authors emphasize that the dark mode of the metasurface prevents radiation into the detection NA, the fact that a measurable scattering spectrum of the bare metasurface has been obtained indicates that residual scattering does exist. This residual signal, even if weak, could act as background noise. Reviewer #2 already noted that the limited sensitivity is likely due to fabrication imperfections, resulting in nonspecific and nonuniform background signals. In their response, the authors acknowledged that improving fabrication quality and array uniformity could further reduce background scattering, which indirectly admits that perfect dark mode suppression is difficult to achieve in practice. However, this limitation is not explicitly described in the main text. It would strengthen the manuscript if the authors could briefly acknowledge this point as a limitation, or provide a short discussion to clarify to what extent intrinsic meta-atom scattering contributes to the background under the dark mode condition.

4. On page 7, line 177, the statement 'In addition, a shorter periodicity with a larger fill factor is preferred for a higher detection yield of the nanoparticles' appears. This raises the question of how such a design fundamentally differs from using a continuous metallic film. Since a bare film would not support the array-induced dark mode that underpins the proposed mechanism, it would be helpful if the authors could clarify this distinction more explicitly in the manuscript, particularly in terms of background suppression, scattering enhancement, and overall detection sensitivity.

5. Fig. 3l shows that the scattering intensity decreases much more slowly with particle size than expected from Rayleigh scattering. The authors attribute this to interference between the particle and the meta-atom array, suggesting an approximate $\sim d^3$ scaling. It remains unclear, however, whether this cubic dependence arises primarily from the general interference mechanism in iSCAT or from a specific particle–meta-atom interaction effect unique to the MAD-iSCAT geometry. Since the particle–meta-atom interaction itself should also be size-dependent, it would strengthen the discussion if the authors could elaborate on how this interaction contributes to the observed scaling behavior. While Figs. 3l and 4j demonstrate weaker-than-Rayleigh scaling down to ~ 20 nm particles, it remains uncertain how far this trend can be maintained for even smaller diameters. In particular, it would be valuable for the authors to discuss whether the scaling slope is expected to remain shallow below 20 nm, and what sets the ultimate detection limit for MAD-iSCAT.

6. In Fig. 5a,b the authors state that exosome sizes are determined through mean-squared displacement (MSD) analysis, with trajectories shown in Supplementary Section 9. However, only sample trajectories are provided, and it is not clear how these lead to MSD values and size estimates. A brief explanation of how MSD is calculated from the trajectories and converted into particle size (e.g., via the diffusion coefficient and Stokes–Einstein relation) would improve clarity.

7. On page 10, line 301, the manuscript states: "This leads to a higher dark mode background level as well as a degraded TP2 mode in silver meta-atoms, which is reflected in the broader wavelength response shown in Fig. 4i compared to the ideal meta-atom resonance." However, Fig. 4i presents the scattering enhancement factor over the visible range, not a resonance broadening effect. This suggests that the figure reference may be incorrect. The authors should verify the figure labeling and clarify which figure actually demonstrates the resonance broadening due to fabrication imperfections—perhaps Fig. 2e?

Overall, the manuscript presents a novel and compelling approach with clear potential impact. I would strongly support its publication once the above clarifications and minor revisions are addressed.

Reviewer #3

(Remarks to the Author)

Review Report:

The article titled, "Meta-amplified dark-field interferometric scattering microscopy" by Lee et al proposes a technique (MAD-iSCAT) that combines dark field microscopy and interference to detect nanoparticles on a plasmonic metasurface. In this report, the authors demonstrated the technique by detecting polymer-nanoparticles and exosomes. They claimed the method to be better than the existing iSCAT technique in terms of sensitivity and contrast. I noted that the first three figures are largely schematic and simulation. The actual results are mainly limited to Fig. 4 and Fig. 5. Moreover, the results are shown on nanoparticles, rather than real systems such as cells, making the work somewhat limited in scope. Here are my detailed comments :

1. One of the major concerns is the lack of validation and poor understanding of what transpires at the metasurface related to field-particle interaction.
2. The particles used by authors are comparatively large. The technique (MAD-iSCAT) needs to demonstrate for small particle (<10 nm nanoparticles and < 300 kDa proteins) detection. This will strengthen the claim experimentally that MAD-iSCAT is better than iSCAT.
3. As claimed, high throughput and sensitivity needs further justification.
4. From the present results, the proposed technique does not appear to offer significant advance compared to iSCAT.
5. The abstract lacks to mention key details / outputs of the MAD-iSCAT technique. This include, (i) radiation modes and why they are special, (2) contrast enhancement, and (3) the level of suppression of the background. These information should have been mentioned in the abstract.
6. The manuscript lacks schematic of detailed optical setup in the main manuscript. This is essential to understand the

proposed technique, instrumentation, and system development.

7. The authors mention, "By careful engineering meta-atom arrays, the presence of nanoscatteer." The statement is vague, needs elaborate explanation, and must be substantiated.

8. In the results section, the authors claim high contrast. This is not substantiated and far from what is evident from the images shown in Fig. 4 and Fig. 5.

9. One of the shortcomings of the present paper is the lack of experimental validation and application on real systems such as cell. Most of the results appear to be simulation based (Fig. 1-3). Actual experimental results are largely limited to Fig. 4 and Fig. 5. It should be other way round i.e., experimental validation must have a larger footprint.

10. In Fig. 5, direct comparison of MAD-iSCAT and iSCAT image is missing.

11. There are many such points that needs to be take care-off in the manuscript.

While the technique appears to be somewhat new, but it does little to advance the field and lacks experimental validation & application. This is in addition to the detailed comments by other two reviewers. The investigation appears to be in its initial stages and requires a lot more conclusive validation to show that MAD-iSCAT is better than iSCAT. Specifically, the images are not convincing and appears to be mild enhancement (visually). Overall, the work may be premature for publication at this stage.

Reviewer #4

(Remarks to the Author)

This manuscript reports the use of a plasmonic metasurface to perform interferometric scattering microscopy (iSCAT). The authors claim that the radiation mode supported by the metasurface interferes with the scattered light from nanoparticles, thereby producing high-contrast interferometric images. Reviewer #1 and Reviewer #2, both experts in iSCAT, have already raised critical questions regarding the novelty and performance of the work, to which the authors have responded. As a specialist in plasmonics and metasurfaces, I will not repeat the iSCAT-related issues already discussed, but instead provide three comments from the perspective of employing metasurfaces for iSCAT.

1. The primary novelty of this work lies in the conceptual innovation of employing a metasurface for iSCAT for the first time. However, the manuscript lacks clarity in its presentation and fails to provide a concise physical picture. As a result, the authors introduce a section, "Principles of contrast amplification in interferometric images," supported by eight equations; this style is more reminiscent of Physical Review rather than Nature Communications. Similarly, the section "The formation of scattering image by metasurface" relies heavily on numerical simulations to justify the mechanism behind, which is not a particularly strong form of evidence.

2. It is well established that plasmonic metasurfaces can enhance light-matter interactions and consequently boost scattering signals. Therefore, the reported contrast enhancement by one to two orders of magnitude (Fig. 4) is in fact expected. However, such enhancement comes at the cost of local heating due to metallic nanostructures. This heating may severely limit the applicability of iSCAT for biologically relevant tasks such as single-molecule tracking. The manuscript does not discuss this critical issue, and such a discussion should be included.

3. Comparative analysis is essential to place the method into context. For example, Table R1-2 in the response letter provides a very useful benchmark. I strongly recommend that the authors incorporate this table into the main text and expand the discussion accordingly.

In summary, the manuscript presents an interesting concept and demonstrates promising experimental results, but the novelty and implications are not clearly articulated. I recommend a major revision before the work can be considered further.

Version 1:

Reviewer comments:

Reviewer #1

(Remarks to the Author)

Thank you for addressing all the comments satisfactorily.

Reviewer #2

(Remarks to the Author)

The manuscript has been significantly improved, and the authors have clearly addressed all of the points I originally raised. I believe it is now of sufficient quality for publication in Nature Communications.

Reviewer #3

(Remarks to the Author)

I prefer to remain anonymous.

Reviewer #4

(Remarks to the Author)

The authors responded very positively and effectively to the issues I mentioned earlier. The current version is satisfactory, and I recommend publication.

Color codes used in this response letter:

Blue: original reviewer comments;

Black: our response;

Red: Modification in the manuscript.

Reviewer #1 (Remarks to the Author):

The manuscript reports of an interferometric technique, MAD-iSCAT, to improve nanoparticle detection sensitivity. Conventional i-SCAT uses a glass slide to accommodate the sample and generate the reference beam, while in MAD-iSCAT this reference beam is generated using a plasmonic metasurface. By engineering the metasurface and the illumination, a dark-field detection scheme is demonstrated. The presence of a particle scatters the nearfield mode of the structure into the far-field. The authors report a modified contrast equation (Eqn. 5,6) because of an additional 'bright mode' arising from the meta-atoms in the presence of a scatterer. As a result, high-contrast images of nanoparticles with enhanced sensitivity over conventional iSCAT is quantified and demonstrated. The results are interesting and the proposed concept is similar to Nature Communications volume 13, 2298 (2022).

(1) The following is not clear from the manuscript with regards to image processing and the contrast equation arising from it: The detected signal in conventional iSCAT in the presence of a scatterer is $I_s = |E_r|^2 + |E_s|^2 + 2|E_s||E_r|\cos\alpha$, where $|E_r|$ is the reference, $|E_s|$ is the scattered and α is the phase difference between these fields. After subtraction of two successive images (with and without a particle) the equation for the detected signal is approximated as $2|E_s||E_r|\cos\alpha$, assuming that $|E_r|$ is constant between the different temporal frames. The contrast of the processed image is $C \approx (2|E_s||E_r|\cos\alpha)/I_{bg}$, where $I_{bg} = |E_r|^2$. In MAD-iSCAT, because of the dark-field detection, ideally no light should reach the camera and hence, this subtraction scheme isn't possible, so $C \approx (|E_r|^2 + 2|E_s||E_r|\cos\alpha)/I_{bg}$. For high-visibility, the magnitudes of the interfering fields must be very close. But since for nanoparticles, $|E_s|$ is low, then $|E_r|$ must also be low if you are to get a good contrast image. So if you assume that $|E_r| \approx |E_s|$ and neglect $|E_r|^2$ or you subtract the background, then the processed image contrast $C \approx (2|E_s||E_r|\cos\alpha)/I_{bg}$, which is the contrast achievable using a conventional iSCAT. However, experimentally there will always be some light reaching the camera, also seen in Fig. 3f. But then again if you perform temporal subtraction, the contrast of the processed image will still be $C \approx (2|E_s||E_r|\cos\alpha)/I_{bg}$.

We thank the reviewer for the thorough review. We agree with the reviewer that with conventional iSCAT, the image contrast remains $C \approx (2|E_s||E_r|\cos\alpha)/I_{bg}$ whether E_r is weak compared to E_s or not. However, the contrast mechanism of MAD-iSCAT is not determined by a constant E_r ; the locally varied E_r' due to the presence of the nanoparticle produces an enhanced scattering mechanism, as we will clarify below.

In both conventional and dark-field iSCAT, subtracting consecutive frames removes the $|E_r|^2$ term and isolates the interference term $2|E_s||E_r|\cos\alpha$, giving rise to a high contrast detection of the weak scattered field of $|E_s|$, which is dependent on E_r . As a strong $|E_r|^2$ can saturate photon detectors, a dark-field iSCAT has an advantage of using the detection dynamic range effectively. While the ideal background intensity I_{bg} is zero in dark-field detection, in practice, any substrate that provides the reference electric field, whether regular glass (Nat. Commun. 13, 2298 (2022)) or the sub-diffraction grating as our plasmonic metasurface, introduces a nonzero I_{bg} that needs to be subtracted. However, despite the better utilization of detection dynamic range, we agree with the reviewer that iSCAT with conventional dark-field processes the same image contrast as standard iSCAT in the form of $C \approx (2|E_s||E_r|\cos\alpha)/I_{bg}$. What makes MAD-iSCAT unique is that the static E_r changes into a bright plasmonic mode E_r' locally due to the perturbation of a nanoparticle nearby and interferes with the scattered field of the nanoparticle. Subtraction of dark-mode frames (without a particle present) removes residual background, leaving only the combined bright-mode and interference signals. Here, the background signal for MAD-iSCAT in a practical experiment is largely governed by the array size and thereby the leakage edge scattering from the boundary of the metasurface, as detailed in the Supplementary Sections 6 and 15. **Unlike earlier dark-field iSCAT, where $|E_r|$ is regarded as constant (always a dark-mode with or without the detection particle), MAD-iSCAT leverages the bright-mode (only with the detection particle) of $|E_r'|$ to boost the scattering signal.**

(a) Please elaborate as to how this Eqn. (5,6) in the manuscript is arrived at and

Thank you for the valuable comments. We have clarified the equations as follows.

The original Equation (5)

$$C = \frac{I_S}{I_{bg}} \approx \frac{2 \sum_{(i,j)} |E_{ij}| |E_s| \cos(\alpha_{ij})}{|\sum_{(i,j)} E_{ij}|^2} \quad (\text{Eq. R1-1})$$

provides a generalized formulation of iSCAT image contrast by decomposing the reference light as a superposition of all fields originating from every position of the detection area. This superposition is crucial to incorporate the local field variation of MAD-iSCAT. In the case of a regular or dark-field iSCAT, the relative phase and amplitude from a glass substrate are fixed, and the superposition $\sum_{(i,j)} E_{ij}$ can be simplified to a static field profile E_r , leading to the well-known simplified form of

$$C = \frac{I_S}{I_{bg}} \approx \frac{2|E_r||E_s|\cos(\alpha)}{|E_r|^2} = \frac{2|E_s|\cos(\alpha)}{|E_r|} \quad (\text{Eq. R1-2})$$

However, MAD-iSCAT simultaneously enhances the numerator as E_{ij} is varying locally depending on the scatter. Exploiting the bright super-scattering plasmon (SSP) modes of our plasmonic meta-atoms, the original Equation (5) is modified for MAD-iSCAT to the original Equation (7) of the manuscript as,

$$C' = \frac{|\sum_{(i,j)} E_{ij}' + E_s|^2 - |\sum_{(i,j)} E_{ij}|^2}{|\sum_{(i,j)} E_{ij}|^2}. \quad (\text{Eq. R1-3})$$

Here, $|\sum_{(i,j)} E_{ij}|^2$ represents the residual background intensity associated with the dark-mode, while E_{ij}' signifies the perturbed field from the bright SSP mode induced by the presence of a particle. The revised MAD-iSCAT contrast formula thus improves both the numerator and denominator. Compared to (Nat. Commun. 13, 2298 (2022)), the numerator gains an altered bright-mode term, instead of relying solely on interfering with the same reference light. Unlike previous approaches that treat the background as a static, noncontributing term, MAD-iSCAT leverages amplitude and phase perturbations in the sub-diffraction-grating metasurface to collect extra photons from the bright mode, resulting in a substantial boost in overall detection signal.

The original Equation (6)

$$SNR = |\sum_{(i,j)} E_{ij}' + E_s| \quad (\text{Eq. R1-4})$$

defines the signal-to-noise ratio (SNR) based on the shot-noise-limited optical detection model, where the noise scales as the square root of the detected photon count, as

$$SNR = \frac{I}{\sqrt{I}} = \sqrt{I}. \quad (\text{Eq. R1-5})$$

In prior iSCAT work (Nat. Commun. 13, 2298 (2022) and others), the SNR is expressed as

$$SNR = \frac{2|E_b||E_s|\cos(\theta)}{\sqrt{|E_b|^2 + 2|E_b||E_s|\cos(\theta) + |E_s|^2}} \approx 2|E_s|\cos(\theta). \quad (\text{Eq. R1-6})$$

For MAD-iSCAT, we accordingly defined it as the original Equation (6)

$$SNR = \frac{|\sum_{(i,j)} E_{ij}' + E_s|^2}{\sqrt{|\sum_{(i,j)} E_{ij}' + E_s|}} = |\sum_{(i,j)} E_{ij}' + E_s|. \quad (\text{Eq. R1-7})$$

We thank the reviewer for the valuable comment. In addressing other reviewer concerns, we have substantially revised those original Equations (5-7) in the manuscript by incorporating a more concrete physical background. Please refer to **Revision 1-2** for the detailed updates.

(b) also why is MAD-iSCAT different from Zhang et.al.s paper cited above? In that work also, light scattered from the coverslip's surface roughness interferes with the scattered light by the sample. It will be beneficial for the readers to clearly understand how you derived the contrast equation, what assumptions are made and how is it different from the above mentioned paper? This will bring the novelty of the paper evident.

Thank you for your insightful question. Here, we clarify the novelty of our work compared to Zhang et al. (Nat. Commun. 13, 2298 (2022)), in which the random surface roughness acts as a reference field that interferes with light scattered by the target particle. In MAD-iSCAT, we demonstrate a novel detection mechanism in which the reference field itself goes through significant perturbation during the detection process by using an artificially engineered metasurface rather than the random roughness to enhance scattering detection. To validate this concept, we have undertaken the following steps.

(1) First, we provide a comprehensive theoretical framework by introducing the sub-diffraction phased array antenna concept to explain how engineered surface features enhance iSCAT detection. This model elucidates why surface roughness serves as an effective reference source and guides the rational design of optimized scattering substrates.

(2) Second, we demonstrate that the proposed super-scattering plasmon (SSP) mode of plasmonic meta-atoms produces much stronger interaction with the nanoparticle compared to a rough glass surface.

As a result, we experimentally confirm that our plasmonic metasurface intensively interacts with the detection object more than the rough surface on the glass. In general, the detection intensity of iSCAT is described as

$$I = |E_r|^2 + 2|E_r||E_s|\cos(\alpha) + |E_s|^2 \quad (\text{Eq. R1-8})$$

When the nanoparticle is large ($|E_r| \ll |E_s|$), the detected intensity scales with the sixth power of the particle diameter as the third term dominates. For smaller particles, where $|E_r| \gg |E_s|$, the second term dominates, and the detection intensity has a cubic power dependence on the particle size. In (Nat. Commun. 13, 2298 (2022)), the cubic power dependence starts to appear for particle sizes smaller than 22 nm. In contrast, MAD-iSCAT shows the approximate cubic power dependence at the particle size over 100 nm and the sixth power dependence is difficult to observe even for the 200-nm particle size. This represents that the plasmonic meta-atom element has much larger scattering cross-sections than the roughness of the bare glass substrate that can interact with the detection object.

Revision 1-1

We appreciate this insightful comment. We have revised the manuscript accordingly to address the reviewer's suggestion.

Main text:

“... The perturbed electric field from meta-atoms generates strong interferometric signals, which exhibit an approximate cubic power law dependence that persists even for comparably larger particle sizes where Rayleigh scattering dominates. ...”

Theoretical framework for contrast and SNR enhancement in MAD-iSCAT

In response to the reviewer's comment, which was also raised by other reviewers, we present a more comprehensive theoretical framework and elucidate the precise mechanism underlying MAD-iSCAT as below.

As noted above, our plasmonic metasurface operates as a phased-array antenna supporting a bright mode. For an ideal metasurface comprising many meta-atoms, the array factor (AF) in the null directions where destructive interference occurs is given by

$$AF_{ideal}(\theta, \varphi, \phi) = \sum_{n=1}^N e^{j\Psi_n} = 0. \quad (\text{Eq. R1-9})$$

When a detection particle resides near a single meta-atom, it introduces an additional phase shift $\Delta\phi$. For a small particle, assuming this perturbation affects only the k -th element in two-dimensional arrays, the modified array factor becomes

$$\begin{aligned} AF_{perturbed}(\theta, \varphi, \phi) &= \sum_{n \neq k} e^{j\Psi_n} + e^{j(\Psi_k + \Delta\phi)} \\ &= AF_{ideal} - e^{j\Psi_k} + e^{j(\Psi_k + \Delta\phi)} = e^{j\Psi_k}(e^{j\Delta\phi} - 1). \end{aligned} \quad (\text{Eq. R1-10})$$

For a given incident intensity I_0 , the scattered power in the bright mode from the phased array, formed by plasmonic meta-atoms each characterized by a single-element scattering cross section σ_{SSP} , is

$$\begin{aligned} P_{BM} &= \sigma_{SSP} I_0 |AF_{perturbed}|^2 \\ &= \sigma_{SSP} I_0 |e^{j\Delta\phi} - 1|^2 = 2\sigma_{SSP} I_0 (1 - \cos(\Delta\phi)). \end{aligned} \quad (\text{Eq. R1-11})$$

For small $\Delta\phi$, applying the small-angle approximation $1 - \cos(\phi) \approx \Delta\phi^2/2$,

$$P_{BM} = \sigma_{SSP} I_0 \Delta\phi^2. \quad (\text{Eq. R1-12})$$

This expression quantifies the scattered power resulting from the perturbed meta-atom element. In practice, the detected interferometric signal arises from the perturbed electric field (from the meta-atom), the scattered field of the particle, and the static background field.^{1, 2} Thus,

$$\begin{aligned}
P_{\text{MAD-iSCAT}} &\propto |E_{BM} + E_s + E_{BG}|^2 \\
&= |E_{BM}|^2 + |E_s|^2 + |E_{BG}|^2 + 2|E_{BM}||E_s|\cos\beta_1 + 2|E_s||E_{BG}|\cos\beta_2 + 2|E_{BG}||E_{BM}|\cos\beta_3
\end{aligned} \tag{Eq. R1-13}$$

where β_1 , β_2 , and β_3 represent the difference in radiation phases.

In conventional iSCAT schemes, the SNR is defined as

$$SNR_{\text{iSCAT}} = \frac{2|E_s||E_{BG}|\cos\beta_2}{\sqrt{|E_s|^2 + 2|E_s||E_{BG}|\cos\beta_2 + |E_{BG}|^2}} \approx 2|E_s|\cos\beta_2. \tag{Eq. R1-14}$$

Similarly, in MAD-iSCAT, with given static background electric fields, the SNR can be expressed as

$$\begin{aligned}
SNR_{\text{MAD-iSCAT}} &= \frac{|E_{BM}|^2 + |E_s|^2 + 2|E_{BM}||E_s|\cos\beta_1 + 2|E_s||E_{BG}|\cos\beta_2 + 2|E_{BG}||E_{BM}|\cos\beta_3}{|E_{BG}|} \\
&\approx SNR_{\text{iSCAT}} + \frac{|E_{BM}|^2 + 2|E_{BM}||E_s|\cos\beta_1 + 2|E_{BG}||E_{BM}|\cos\beta_3}{|E_{BG}|}.
\end{aligned} \tag{Eq. R1-15}$$

The contrast of MAD-iSCAT can be described as

$$\begin{aligned}
C_{\text{MAD-iSCAT}} &= \frac{|E_{BM}|^2 + |E_s|^2 + 2|E_{BM}||E_s|\cos\beta_1 + 2|E_s||E_{BG}|\cos\beta_2 + 2|E_{BG}||E_{BM}|\cos\beta_3}{|E_{BG}|^2} \\
&\approx C_{\text{iSCAT}} + \frac{|E_{BM}|^2 + 2|E_{BM}||E_s|\cos\beta_1 + 2|E_{BG}||E_{BM}|\cos\beta_3}{|E_{BG}|^2}.
\end{aligned} \tag{Eq. R1-16}$$

This demonstrates that compared to previous dark-field iSCAT configurations, where the interfacing glass surface often presents random features, MAD-iSCAT enables significantly enhanced light signals for detection.

We greatly appreciate the reviewer's insightful comments, which have enabled us to deepen the theoretical foundation of MAD-iSCAT.

Revision 1-2

In line with another reviewer's suggestion for a more concise presentation, we now provide detailed explanations together with equations in the Supplementary Information.

Main text:

“... It has been shown that dark-field images are also generated through the interference between scattered lights from the object and surface roughness (**static background**). ...”

“... where E_{ij} and E_s are the electric field distribution generated by the meta-atom elements at $(x_{ij}, y_{ij}, -z_a)$ and the scatterer at $(0, 0, z_s)$, respectively. α_{ij} denotes the difference in radiation phases between the meta-atom elements and the scatterer. For deep-subwavelength scatterers, the detected scattering intensity $|E_s|^2$ scales with the **sixth power of the particle size in conventional dark-field imaging (Fig. 1c)**. In previous regular and dark-field iSCAT schemes, E_{ij} terms in Equation (2) are regarded as a static random background. Especially, $|\sum_{(i,j)} E_{ij}|^2$ is not involved in the scattered light detection and is removed by subtracting consecutive frames.

In this study, we introduce a metasurface comprising subwavelength meta-atom arrays to improve the detection sensitivity in iSCAT. Each metallic meta-atom possesses a large σ_{sc} compared to the dielectric nanoparticle (Fig. 1d). Without a nanoscatterer, the meta-atom array behaves as a sub-diffraction grating, primarily reflecting incident light (constructive interference). In contrast, the excited scattering modes of the meta-atoms collectively create destructive interference towards the free space within the detection numerical aperture (NA), resulting in an imaginary Poynting vector (Fig. 1e). This destructive interference, termed dark mode of the metasurface, is represented by $|\sum_{(i,j)} E_{ij}|^2$ term in Equation (2) in the absence of a nanoscatterer. When a nanoscatterer approaches, it disturbs the local resonance mode of the meta-atom elements and alters the phase profile of the sub-diffraction grating (Fig. 1f). The design of the meta-atom is chosen at a resonance mode, such that the amplitude A_{ij} and phase ϕ_{ij} of the resonance mode are sensitive to the nearby environmental change due to the presence of the nanoscatterer, and the perturbed mode within the sub-diffraction array radiates towards the detection optics. Considering the perturbed electric fields of meta-atom elements, Equation (2) is modified as

$$|\sum_{(i,j)} E_{ij}'|^2 + 2 \sum_{(i,j)} |E_{ij}'| |E_s| \cos(\alpha_{ij}) + |E_s|^2. \quad (3)$$

The bright mode of the metasurface $|\sum_{(i,j)} E_{ij}'|^2$ and $|E_{ij}'|$ thereby generates an interferometric image with significantly enhanced contrast and SNR. Accordingly, the image contrast for MAD-iSCAT can be written as

$$C' = \frac{|\sum_{(i,j)} E_{ij}' + E_s|^2 - |\sum_{(i,j)} E_{ij}|^2}{|\sum_{(i,j)} E_{ij}'|^2}. \quad (4)$$

We implement such a plasmonic metasurface with sub-diffraction hexagonal meta-atom arrays, where degenerate plasmonic hybridized modes achieve a strong scattering condition (see details in Supplementary Section 1). ...”

Supplementary information:

“**Supplementary Section 1.** Detailed explanation of the working mechanism of MAD-iSCAT.

Our plasmonic metasurface operates as a phased-array antenna supporting a bright mode. For an ideal metasurface comprising many meta-atoms, the array factor (AF) in the null directions where destructive interference occurs is given by

$$AF_{ideal}(\theta, \varphi, \phi) = \sum_{n=1}^N e^{j\Psi_n} = 0. \quad (\text{Eq. S1})$$

When a detection particle resides near a single meta-atom, it introduces an additional phase shift $\Delta\phi$. Assuming this perturbation affects only the k -th element in two-dimensional arrays, the modified array factor becomes

$$\begin{aligned} AF_{perturbed}(\theta, \varphi, \phi) &= \sum_{n \neq k} e^{j\Psi_n} + e^{j(\Psi_k + \Delta\phi)} \\ &= AF_{ideal} - e^{j\Psi_k} + e^{j(\Psi_k + \Delta\phi)} = e^{j\Psi_k} (e^{j\Delta\phi} - 1). \end{aligned} \quad (\text{Eq. S2})$$

For a given incident intensity I_0 , the scattered power in the bright mode from the phased array, formed by plasmonic meta-atoms each characterized by a single-element scattering cross section σ_{SSP} , is

$$\begin{aligned} P_{BM} &= \sigma_{SSP} I_0 |AF_{perturbed}|^2 \\ &= \sigma_{SSP} I_0 |e^{i\Delta\phi} - 1|^2 = 2\sigma_{SSP} I_0 (1 - \cos(\Delta\phi)). \end{aligned} \quad (\text{Eq. S3})$$

For small $\Delta\phi$, applying the small-angle approximation $1 - \cos(\phi) \approx \Delta\phi^2/2$,

$$P_{BM} = \sigma_{SSP} I_0 \Delta\phi^2. \quad (\text{Eq. S4})$$

This expresses the scattered power resulting from the perturbed meta-atom element. In practice, the detected interferometric signal arises from the superposition of the perturbed electric field (meta-atom), the scattered field of the particle, and the static background field.^{1, 2} Thus,

$$\begin{aligned} P_{MAD-iSCAT} &\propto |E_{BM} + E_s + E_{BG}|^2 \\ &= |E_{BM}|^2 + |E_s|^2 + |E_{BG}|^2 + 2|E_{BM}||E_s|\cos\beta_1 + 2|E_s||E_{BG}|\cos\beta_2 + 2|E_{BG}||E_{BM}|\cos\beta_3 \end{aligned}$$

where β_1 , β_2 , and β_3 represent the difference in radiation phases. The size dependency of $|E_{BM}|^2$ is more complex than the simple cubic or sixth-power scaling characteristic of conventional iSCAT or pure Rayleigh scattering. The maximum power dependence of $|E_{BM}|^2$ follows a sixth-power law. Thus, for small nanoparticles, the power dependency tends to approximate a near-cubic scaling when the interference terms are dominant.

In conventional iSCAT schemes, the SNR is defined as

$$SNR_{iSCAT} = \frac{2|E_s||E_{BG}|\cos\beta_2}{\sqrt{|E_s|^2 + 2|E_s||E_{BG}|\cos\beta_2 + |E_{BG}|^2}} \approx 2|E_s|\cos\beta_2. \quad (\text{Eq. S5})$$

Similarly, in MAD-iSCAT, with given static background electric fields, the SNR can be expressed as

$$\begin{aligned} SNR_{MAD-iSCAT} &= \frac{|E_{BM}|^2 + |E_s|^2 + 2|E_{BM}||E_s|\cos\beta_1 + 2|E_s||E_{BG}|\cos\beta_2 + 2|E_{BG}||E_{BM}|\cos\beta_3}{|E_{BG}|} \\ &\approx SNR_{iSCAT} + \frac{|E_{BM}|^2 + 2|E_{BM}||E_s|\cos\beta_1 + 2|E_{BG}||E_{BM}|\cos\beta_3}{|E_{BG}|}. \end{aligned} \quad (\text{Eq. S6})$$

The contrast of MAD-iSCAT can be described as

$$\begin{aligned} C_{MAD-iSCAT} &= \frac{|E_{BM}|^2 + |E_s|^2 + 2|E_{BM}||E_s|\cos\beta_1 + 2|E_s||E_{BG}|\cos\beta_2 + 2|E_{BG}||E_{BM}|\cos\beta_3}{|E_{BG}|^2} \\ &\approx C_{iSCAT} + \frac{|E_{BM}|^2 + 2|E_{BM}||E_s|\cos\beta_1 + 2|E_{BG}||E_{BM}|\cos\beta_3}{|E_{BG}|^2}. \end{aligned} \quad (\text{Eq. S7})$$

This demonstrates that compared to previous dark-field iSCAT configurations, where the interfacing glass surface often presents random features, MAD-iSCAT enables significantly enhanced light signals for detection.”

(c) When will this assumption break? For e.g., the plot in Fig. 3I, when will they intersect? Of course, iSCAT is for the detection of nanoparticles. So this question is purely out of curiosity to see when these assumptions break.

In reality, once particle diameters exceed ~200 nm, the pure Rayleigh scattering transitions toward the more complex Mie scattering regime. Consequently, scattering intensity no longer follows a strict sixth-power law for larger particles. MAD-iSCAT, however, combines contributions of direct particle scattering, the bright mode, and the static background. So its signal curve does not intersect that of either pure Rayleigh or Mie scattering. To illustrate this, we have replaced the Rayleigh scattering cross-section in our analysis with the appropriate Mie scattering cross-section for particles up to 800 nm. Within this size range, Rayleigh and Mie predictions remain similar, but above 200 nm the size exponent falls below six, as shown in Figure R1-1. Thus, even though bright-mode enhancement diminishes for very large particles, MAD-iSCAT still provides an elevated signal relative to pure scattering, and no intersection appears in Fig. 3I.

Figure R1-1. Mie scattering cross-section of a polystyrene bead.

Revision 1-3

We appreciate this comment to clarify and enhance the manuscript. In line with another reviewer's suggestion for footprint, we revised the Fig. 3I to Supplementary Information.

Supplementary information:

“Supplementary Section 8. Amplitude and phase perturbations of the meta-atom in the presence of a detection target.

Supplementary Fig. 7. a,b, The changes of (a) σ_{sc} and (b) phase of a meta-atom element with polystyrene (PS) nanospheres having different diameters when the array size is $9.6 \times 9.3 \mu\text{m}^2$. The inset shows the meta-atom element interfacing with the detection target atop. **c,** The comparison of the scattering intensities from MAD-iSCAT and Mie scattering cross-section.”

(2) Lines 54-55: “However, as the light scattering cross-section scales with the sixth power of the particle’s diameter...”

(a) Rayleigh scattering scales with sixth power of the particle radius and not diameter. Please check and correct.

Thank you for this observation. Rayleigh scattering scales strictly with the sixth power of the particle’s radius r , i.e., $\sigma \propto r^6$, or when using diameters, $\sigma \propto 1/64 d^6$ with a constant scaling factor of $1/64$. The scattering size dependency remains the same for either particle size metric. We have chosen to express size in terms of diameter throughout the manuscript for consistency with other iSCAT studies.

(b) Fig. 4l – black dashed line represents cubic power. Is this plotted as a function of radius or diameter?

Thank you for pointing this out. The black dashed line in Fig. 4l is plotted against diameter. For Rayleigh scattering, intensity scales as the sixth power of the diameter, whereas in iSCAT the detected signal follows a cubic dependence on diameter, so we have used diameter consistently for both cases. In revision, Fig. 4l was moved to Fig. 5c.

(c) Fig. 4j – gray dashed line represents sixth power dependence.

On the PMMA substrate in air, the scattering intensity of particles follows a sixth-power dependence on diameter. The gray dashed line in Fig. 4j is plotted accordingly for comparison.

(d) Explain Fig. 4l and Fig. 4j based on Rayleigh scattering scaling inversely as sixth power of radius. Please mention the values of the scattering intensity at 50 nm, 100 nm, upto 200 nm for both MAD-iSCAT and Rayleigh scattering. It is provided in the plot, but it is easier to compute the dependence on particle size from numbers directly given.

Thank you for this insightful suggestion. On a PMMA substrate in air, particle scattering intensity follows a sixth-power scaling with diameter, which the gray dashed line in Fig. 4j confirms. Because absolute cross-sections are commonly challenging to measure, we instead report relative scattering intensities using standard polystyrene beads. As requested, we have added values of scattering intensity at diameters in Fig. 4j in the Supplementary Information. The comparative values for MAD-iSCAT and regular iSCAT are already provided in the main text (first paragraph of the Experimental Results section). In Fig. 4l, iSCAT is dominated by the interference term $2|E_s||E_r|$, which scales as the cube of particle diameter, rather than the Rayleigh $|E_s|^2$ term. The iSCAT data follow this cubic dependence, and MAD-iSCAT likewise exhibits a power dependence near the cubic law power. In revision, Fig. 4l was moved to Fig. 5c.

Revision 1-4

We appreciate this comment to clarify and enhance the manuscript. We have added values of scattering intensity at diameters in Fig. 4j to Supplementary Information.

Supplementary information:

“Supplementary Section 12. The comparison of MAD-iSCAT and Rayleigh scattering.

Supplementary Table 1. Scattering intensities of polystyrene beads used in Fig. 4j.

Size (nm)	MAD-iSCAT		Rayleigh scattering	
	Mean (a.u.)	Standard deviation (a.u.)	Mean (a.u.)	Standard deviation (a.u.)
46	0.109	0.0312	0.000759	N/A
60	0.235	0.0534	0.00317	N/A
100	1.036	0.3216	0.0495	0.0100
200	12.74	2.1802	2.302	0.2517

For the Rayleigh scattering plot of 46- and 60-nm polystyrene beads, a size-dependent power law of 5.38 was applied.² In this size regime, background signals dominate and direct difference imaging in an air environment with drop-cast beads on bare surfaces is not practical.”

(3) Fig. 4k --- can you explain why the particle locations have negative pixel values? Isn't particle location supposed to be affecting the pixel value because of Gouy phase? Then why do they have positive values in the Mad-iSCAT?

The Gouy phase shift introduces a π phase change for light traversing the focal region. In conventional iSCAT, only the scattered field from the particle acquires this Gouy phase shift, whereas the reference reflection does not. This phase difference underlies the axial contrast in iSCAT. When the particle's scattered light is π out of phase with the reflected field, destructive interference produces a darker signal, yielding negative pixel values relative to the background. In contrast, MAD-iSCAT employs dark-field interferometric illumination rather than relying on a plane-wave reflection. Because the reference field is generated by the bright super-scattering plasmon mode and not by a focused reflection, the scattered signal appears brighter than the surrounding background when the particle is within the focal plane. Consequently, particle locations in MAD-iSCAT manifest as positive pixel values.

(4) Fig. 4f – it is mentioned that all particles are validated with ground truth fluorescence and SEM images.

(a) Is this done for all the results shown in the manuscript?

The ground-truth fluorescence and SEM validations were performed primarily for the data in Fig. 3a–j to confirm that each optical scattering signal corresponds to an individual nanoparticle rather than an aggregate and to verify its spectral response at a known location under air conditions. For experiments conducted in aqueous environments, such as those involving 80, 40, and 20 nm polystyrene beads, exosomes, and ferritin, fluorescence bleaching and SEM fixation are impractical. In these cases, we rely on statistical sampling, acquiring a larger number of detection events, consistent with standard practice in iSCAT studies of particles in liquid.

(b) How was it ensured that fluorescence signal didn't leak into the iSCAT measurements?

Thank you for your valuable comments. Prior to the fluorescence-bead experiments, we applied maximum laser power at the fluorophore excitation wavelength for several hours to bleach the beads' fluorescence emission. We have also confirmed that the scattering intensity is much stronger than the fluorescence emission.

Revision 1-5

We appreciate this comment to clarify and enhance the manuscript. We have revised the manuscript accordingly to address the reviewer's concerns.

Methods:

“... Fluorescence beads are bleached before the scattering measurements. ...”

(5) Since MAD-iSCAT is a near-field technique, it will be good to have a plot illustrating the field strength as a function of distance from the meta-atom surface.

We appreciate this comment to clarify and enhance the manuscript. The field distribution along the z-axis is shown in Fig. R1-2.

Figure R1-2. a-c, Field intensity distribution within a rectangular unit cell along the z-axis at heights of (a) 5 nm, (b) 15 nm, and (c) 25 nm.

Revision 1-6

We have added the suggested simulation results, as shown in Figure R1-2, in the Supplementary Information.

Methods:

“... The near-field distributions are shown in Supplementary Section 16. ...”

Supplementary information:

“Supplementary Section 16. Near-field distribution.”

Supplementary Fig. 13. a-c, Field intensity distribution within a rectangular unit cell along the z-axis at heights of (a) 5 nm, (b) 15 nm, and (c) 25 nm.”

(6) It was not clear whether the SSP mode was utilized for all the experimental results demonstrated in the manuscript.

Thank you for this insightful comment. We employed the SSP mode of our plasmonic metasurface in all experimental measurements. Embedding the meta-atoms within a PMMA layer ensures that the SSP resonance remains unchanged despite variations in

the overlying medium. Because the refractive index of PMMA is constant, we adjust the incident angle to satisfy the same plasmon modes excitation, 75° in air and 46° in aqueous environments. In both cases, the in-plane wavevector in PMMA satisfies $k_x = k_0 \sin 75^\circ = k_{\text{water}} \sin 46^\circ = k_{\text{PMMA}} \sin 40^\circ$. The SSP mode's field profiles and scattering cross-sections are approximately identical under these conditions.

Revision 1-7

We appreciate this comment to clarify and enhance the manuscript. We have revised the manuscript accordingly to address the reviewer's concerns.

Methods:

“... Incident angles are selected to satisfy the same plasmon modes excitation with $k_x = k_0 \sin 75^\circ = k_{\text{water}} \sin 47^\circ = k_{\text{PMMA}} \sin 40^\circ$”

(7) Fig. 2a, can you provide the simulation results with and without the nano scatterer. This will aid the reader to see how the scatterer affects the confined mode.

We appreciate this comment to clarify and enhance the manuscript. The simulation results with and without the nanosscatterer are shown in Fig. R1-3.

Figure R1-3. a,b, Exemplary scattered electric field distributions (a) without a 200-nm polystyrene nanosscatterer and (b) with a polystyrene nanosscatterer.

Revision 1-8

We appreciate this comment to clarify and enhance the manuscript. We have added the suggested simulation results, both with and without the nanoscale scatterer, as shown in Figure R1-3, in the Supplementary Information.

Supplementary information:

“Supplementary Section 3. Exemplary scattered electric field distributions with a polystyrene nanosscatterer.

Supplementary Fig. 2. a,b, Exemplary scattered electric field distributions (a) without a 200-nm polystyrene nanosscatterer and (b) with a polystyrene nanosscatterer.”

(8) Fig.3 - What is the spatial extent covered by the polar plots with respect to the meta-atom array? Maybe add a scale bar to highlight it and also, where exactly on the meta-atom array is this measurement carried out? Is it at the intersection of the dotted lines in Fig. 3a?

Thank you for your detailed review. Using multipole decomposition and the Green’s tensor formalism, we computed the far-field radiation patterns. The polar plots in Fig. 3 span an arbitrary angular range in the far field. Specifically, Fig. 3b,e,h depict the upper hemisphere (reflection) of the far field, with a hemisphere radius set to 2 mm. The metasurface array is positioned at the center of the hemisphere’s base plane. A polar plot of the far-field radiation pattern was generated from meta-atom arrays ($19.4 \times 19.3 \mu\text{m}^2$).

Revision 1-9

We appreciate this comment to clarify and enhance the manuscript. We have revised the manuscript accordingly to address the reviewer's concerns.

Methods:

“... The far-field radiation pattern is calculated at a distance of 2 mm from the array center. ...”

(9) Fig.1(c-f) – a suggestion is that the plots can be better represented. For example, wavelength range along the x-axis will be useful as these meta-atoms do not operate at all wavelengths. Another point of confusion was the grey plot in the case of the dark-mode. This can be understood after reading the manuscript, but a bit more detailed explanation in the figure caption will also help.

Revision 1-10

We appreciate this comment to clarify and enhance the manuscript. We have revised Fig. 1 accordingly to address the reviewer's concerns.

Main text:

(10) Though the array spans a few microns, what is the field-of-view achievable (FoV)? Is it $9.6 \times 9.3 \mu\text{m}^2$? Do you think you can enhance the FoV? It will be useful to highlight these issues in the manuscript.

Thank you for your detailed review. The achievable field of view is primarily limited by fabrication capabilities, but larger areas are technically accessible. We chose a 40×40 grid of $9.6 \times 9.3 \mu\text{m}^2$ meta-atom arrays to facilitate SEM correlation by identifying the specific optical array before locating it under SEM. For future work such as cellular imaging, which requires areas larger than $10 \times 10 \mu\text{m}^2$, we anticipate scaling the FOV to at least $19.4 \times 19.3 \mu\text{m}^2$ or even up to $48 \times 46.15 \mu\text{m}^2$. Larger arrays will accommodate typical cell sizes and further suppress background intensity, as demonstrated in our Supplementary Section 6.

Revision 1-11

We appreciate this comment to clarify and enhance the manuscript. We have revised the manuscript accordingly to address the reviewer's concerns.

Main text:

“... MAD-iSCAT has the potential to dramatically improve single-molecule mass imaging throughput by orders of magnitude through enhanced fabrication approaches such as self-assembly for better metasurface quality⁵¹ and large array geometries to enable high-throughput nanoparticle imaging with expanded field-of-view and suppressed background noise. ...”

(11) MAD-iSCAT is a near-field detection technique. Conventional iSCAT can help isolate particles within the focal volume of the detection objective. Therefore, though it is obvious, highlighting the limitations of MAD-iSCAT is important especially for biologists.

We appreciate you highlighting this important consideration. Many iSCAT applications achieve exceptional sensitivity for small-nanoparticle detection using wide-field imaging, without the need to adjust the focal plane extensively above the glass–air interface. MAD-iSCAT, however, operates within the evanescent-field regime (≤ 200 nm from the surface). Thus, for routine nanoparticle sensing where long-range 3D tracking is unnecessary, MAD-iSCAT is fully compatible with conventional iSCAT methods. In contrast, biological cell imaging will be limited to a very thin axial region near the substrate.

Revision 1-12

We appreciate this comment to clarify and enhance the manuscript. We have revised the manuscript accordingly to address the reviewer's concerns.

Main text:

“... It is important to note that the current MAD-iSCAT has a detection volume more constrained than conventional iSCAT and suffers from fabrication imperfections, array inhomogeneities, and

finite-size effects. Nonetheless, MAD-iSCAT combines the advantages of iSCAT and meta-atom arrays, opening new opportunities for optimization in label-free sensing applications. ...”

(12) Fig. 4j – please correct the legends in the plot.

Revision 1-13

We appreciate this comment to clarify and enhance the manuscript. We have revised the manuscript accordingly to address the reviewer's concerns.

Main text:

(13) Due to enhanced field strengths, can particles get trapped to the high field intensity?

We appreciate this observation. Optical trapping relies on the gradient of the electromagnetic field intensity, drawing particles toward regions of highest field enhancement. Achieving sufficient trapping stiffness for nanoparticles against Brownian

motion typically requires either very high laser power densities or tightly confined fields within nanoscale gaps. In our design, the plasmonic meta-atoms are embedded beneath a PMMA layer, which attenuates the field enhancement at the surface where particles reside. With a 145 nm lattice period and no sharp gap structures, the field enhancement at the PMMA interface remains moderate (under $\sim 20\times$), and MAD-iSCAT contrast arises primarily from interferometric detection rather than extreme field confinement. Consequently, we do not expect significant optical trapping or thermophoretic effects under our operating conditions.³

(14) Have you quantified any temperature changes due to plasmonic effects? Will this affect the biological samples under study?

Thank you for highlighting this important factor. We used a relatively low laser power density of 42 W/cm². In previous studies of silver nanospheres (10–50 nm radius) at 4,000 W/cm², where scattering cross-sections peaked as 1×10^{-14} m², temperature rises of 0.5–3 °C were confirmed. Since our SSP mode features cross-sections in the same order of magnitude but operates at 100× lower power density, we estimate a temperature increase of less than 1 °C. Under these conditions, biological samples are not affected by plasmonic heating, alleviating concerns about thermal damage.⁴

Revision 1-14

We appreciate this observation. We have revised the manuscript accordingly to address the reviewer's concerns.

Main text:

“... The laser power employed is sufficiently low to avoid significant photothermal effects and our protective layer also minimizes the heating effect on detection targets and prevents photodamage to silver meta-atoms.⁵¹ ...”

In conclusion, we sincerely appreciate the reviewer's detailed evaluation and valuable comments that have enhanced our manuscript.

Reviewer #2 (Remarks to the Author):

The manuscript entitled “Meta-amplified dark-field interferometric scattering microscopy” has undergone one round of peer review, and the authors have revised the manuscript in response to the reviewers’ comments. After reviewing the revised version and the response letter, I find that the authors have addressed the earlier concerns in a satisfactory manner.

A notable strength of this work lies in the central concept of exploiting the dark mode of the plasmonic meta-atom array. By engineering the metasurface so that its collective radiation is suppressed in the absence of objects, the authors achieve a background-free baseline. Upon the attachment of nanoparticles, local perturbations induce a transition from the dark mode to a bright radiation mode, resulting in a strong and detectable signal. This mechanism represents an original idea that clearly differentiates MAD-iSCAT from previously reported iSCAT approaches. The physical picture is compelling and provides a solid foundation for the reported contrast enhancements. I strongly recommend this work for publication after one further round of minor revisions.

1. On page 3, line 57, the statement ‘Interferometric imaging addresses this challenge by leveraging the interference of scattered light from nanoscale objects with reference light, yielding label-free images with high sensitivity and resolution’ seems to overstate the claim of ‘high resolution’. Interferometric imaging does not inherently surpass the diffraction limit; rather, it improves localization precision or effective resolution through enhanced signal-to-noise.

Revision 2-1

We appreciate this comment to clarify and enhance the manuscript. We have revised the manuscript accordingly to address the reviewer’s concerns.

Main text:

“... Interferometric imaging addresses this challenge by leveraging the interference of scattered light from nanoscale objects with reference light, yielding label-free images with high sensitivity. ...”

2. In Fig. 1a, the schematic illustrates a reflection geometry where the illumination incidence angle is drawn larger than the detection NA cone, suggesting complete separation between illumination and detection. However, for a 0.9 NA dark-field objective, the illumination annulus specified by the manufacturer is typically $NA = 0.8-0.9$, corresponding to $\sim 53^\circ-64^\circ$ in air, which partially overlaps with the collection cone ($0-64^\circ$). This discrepancy between the schematic, the stated 75° incidence angle, and the actual objective specification should be clarified. In

principle, one could increase the laser input incidence angle beyond the nominal annulus by back focal plane engineering while restricting the detection NA to a smaller cone. If such a strategy was employed, it would be important for the authors to explicitly state this in the Methods, as it would resolve the apparent inconsistency.

Thank you for your detailed review. As shown in Figure R2-1, the dark-field objective lens (UMPlanFI, 100 \times , 0.9 NA, Olympus) supports an oblique hollow light cone illumination. The incident angle of the dark-field objective lens is measured to be 75 $^\circ$.

Figure R2-1. An oblique hollow light cone illumination of a dark-field objective.

Revision 2-2

We have added the information related to Figure R2-1 in the Supplementary Information.

Supplementary Information:

“**Supplementary Section 10. MAD-iSCAT imaging in air environments.**”

Supplementary Fig. 9. An oblique hollow light cone illumination of a dark-field objective. ”

3. Despite the conceptual novelty, one potential weakness of the approach is that achieving a perfect dark mode in practice may be difficult. Fabrication imperfections, array inhomogeneities, and finite-size effects could lead to residual scattering from the meta-atoms themselves, limiting the reproducibility and robustness of the method. It would be helpful if the authors could briefly acknowledge this limitation in the manuscript, as it provides important context for the practical implementation of MAD-iSCAT. Although the authors emphasize that the dark mode of the metasurface prevents radiation into the detection NA, the fact that a measurable scattering spectrum of the bare metasurface has been obtained indicates that residual scattering does exist. This residual signal, even if weak, could act as background noise. Reviewer #2 already noted that the limited sensitivity is likely due to fabrication imperfections, resulting in nonspecific and nonuniform background signals. In their response, the authors acknowledged that improving fabrication quality and array uniformity could further reduce background scattering, which indirectly admits that perfect dark mode suppression is difficult to achieve in practice. However, this limitation is not explicitly described in the main text. It would strengthen the manuscript if the authors could briefly acknowledge this point as a limitation, or provide a short discussion to clarify to what extent intrinsic meta-atom scattering contributes to the background under the dark mode condition.

We appreciate this observation. Thank you for highlighting this important factor. Although the residual scattering helps to obtain an interferometric image,^{1,2} we acknowledge that current MAD-iSCAT obviously shows a larger background level than the dark-field iSCAT with continuous substrate.

Revision 2-3

We appreciate this comment to clarify and enhance the manuscript. We have revised the manuscript accordingly to address the reviewer's concerns.

Main text:

“... It is important to note that the current MAD-iSCAT has a detection volume more constrained than conventional iSCAT and suffers from fabrication imperfections, array inhomogeneities, and finite-size effects. Nonetheless, MAD-iSCAT combines the advantages of iSCAT and meta-atom arrays, opening new opportunities for optimization in label-free sensing applications. MAD-iSCAT has the potential to dramatically improve single-molecule mass imaging throughput by orders of magnitude through enhanced fabrication approaches such as self-assembly for better metasurface quality and large array geometries to enable high-throughput nanoparticle imaging with expanded field-of-view and suppressed background noise. ...”

4. On page 7, line 177, the statement ‘In addition, a shorter periodicity with a larger fill factor is preferred for a higher detection yield of the nanoparticles’ appears. This raises the question of how such a design fundamentally differs from using a continuous metallic film. Since a bare film would not support the array-induced dark mode that underpins the proposed mechanism, it would be helpful if the authors could clarify this distinction more explicitly in the manuscript, particularly in terms of background suppression, scattering enhancement, and overall detection sensitivity.

Thank you for your insightful question. A similar approach using bare gold films was demonstrated by Zhang et al. (Nat. Methods 17, 1010 (2020)). While continuous films rely on surface roughness to provide a reference field for interferometric imaging, we demonstrate that these naturally occurring irregularities can be artificially engineered into periodic meta-atoms to enhance scattering detection. A key difference between continuous metallic film and the MAD-iSCAT is that the metasurface is designed to take advantage of the SSP mode of the meta-atoms, and therefore creates much stronger scattering interactions. The periodicity of the metasurface is chosen so that the resonance mode provides maximized coverage of the entire patterned area, while still maintaining the SSP mode for the meta-atoms, as shown in Figure 2d.

As the reviewer correctly noted, continuous metallic films offer superior background suppression. In MAD-iSCAT, background intensity arises from two primary factors: finite array size and fabrication imperfections. As shown in our Supplementary Section 6, larger arrays minimize background scattering at the center by reducing edge diffraction, while

fabrication errors also contribute unwanted signals. These results demonstrate that array size significantly affects background suppression. We anticipate scaling our arrays to at least $19.4 \times 19.3 \mu\text{m}^2$ or up to $48 \times 46.15 \mu\text{m}^2$ for future applications.

In general, the detection intensity of iSCAT is described as

$$I = |E_r|^2 + 2|E_r||E_s|\cos(\alpha) + |E_s|^2. \quad (\text{Eq. R2-1})$$

The detection intensity has a cubic power dependence on the particle size since the second term dominates over the third term. However, when $|E_r| < |E_s|$ the intensity becomes proportional to the sixth power of the particle size (the third term becomes dominant). In Zhang et al. (Nat. Methods 17, 1010 (2020)), the cubic power dependence starts to appear for particle sizes smaller than 100 nm. In contrast, MAD-iSCAT shows approximate cubic dependence for particles over 100 nm, with sixth-power scaling difficult to observe even at 200 nm diameter. This demonstrates that our plasmonic meta-atoms provide much larger scattering cross-sections (which can transition between dark and bright modes) than bare gold film roughness, offering enhanced sensitivity for smaller nanoparticles. This represents the key novelty of our approach.

Revision 2-4

We appreciate this insightful comment. We have revised the manuscript accordingly to address the reviewer's suggestion.

Main text:

“... The perturbed electric field from meta-atoms generates strong interferometric signals, which exhibit an approximate cubic power law dependence that persists even for comparably larger particle sizes where Rayleigh scattering dominates. ...”

5. Fig. 3l shows that the scattering intensity decreases much more slowly with particle size than expected from Rayleigh scattering. The authors attribute this to interference between the particle and the meta-atom array, suggesting an approximate $\sim d^3$ scaling. It remains unclear, however, whether this cubic dependence arises primarily from the general interference mechanism in iSCAT or from a specific particle–meta-atom interaction effect unique to the MAD-iSCAT geometry. Since the particle–meta-atom interaction itself should also be size-dependent, it would strengthen the discussion if the authors could elaborate on how this interaction contributes to the observed scaling behavior. While Figs. 3l and 4j demonstrate weaker-than-Rayleigh scaling down to ~ 20 nm particles, it remains uncertain how far this trend can be maintained for even smaller diameters. In particular, it would be valuable for the authors to discuss whether the scaling slope is expected to remain shallow below 20 nm, and what sets the ultimate detection limit for MAD-iSCAT.

Thanks to the reviewer for the valuable comment and insightful observation. We acknowledge that explicit information regarding the size dependence of MAD-iSCAT was not clearly provided in the original manuscript. To address this point, we have developed a more comprehensive theoretical framework to clarify the underlying mechanism of MAD-iSCAT. Since this issue was also raised by other reviewers and discussed in detail over four pages, we kindly refer the reviewer to our previous response in the section **Theoretical framework for contrast and SNR enhancement in MAD-iSCAT** on page 5. In brief, the size dependence of light scattering mediated by the bright mode in our metasurface is more complex than the simple cubic or sixth-power scaling typically observed in conventional iSCAT or Rayleigh scattering. For small nanoparticles, the power dependence approaches a near-cubic scaling regime when interference terms dominate the response.

Revision 2-5

We have revised the manuscript accordingly to address the reviewer's concern. Please kindly refer to the **Revision 1-2** for detailed updates.

6. In Fig. 5a,b the authors state that exosome sizes are determined through mean-squared displacement (MSD) analysis, with trajectories shown in Supplementary Section 9. However, only sample trajectories are provided, and it is not clear how these lead to MSD values and size estimates. A brief explanation of how MSD is calculated from the trajectories and converted into particle size (e.g., via the diffusion coefficient and Stokes–Einstein relation) would improve clarity.

Thank you for your detailed review. To perform MSD analysis, particle trajectories $r(t) = (x(t), y(t))$ are constructed by recording positions across N frames with an exposure time of Δt . At each time lag τ , the MSD is calculated by averaging the squared displacements between all position pairs separated by this specific lag interval as

$$MSD(\tau) = \langle |r(t + \tau) - r(t)|^2 \rangle \quad (\text{Eq. R2-2})$$

where the angular brackets denote ensemble averaging over all applicable time points t . For exosome characterization, high-speed imaging is conducted with an exposure time of 487.218 μs to capture rapid Brownian motion, with individual exosome trajectories extracted as shown in the Supplementary Fig. 11. When analyzing MSD as a function of time lag τ , freely diffusing particles exhibiting Brownian motion demonstrate a linear relationship described by

$$MSD(\tau) = error + b\tau \quad (\text{Eq. R2-3})$$

where the intercept term corrects for localization errors and the slope directly scales with the diffusion coefficient. In the case of three-dimensional Brownian motion projected onto

a two-dimensional imaging plane, the coefficient is given by $D = b/4$. Finally, the exosome's radius r is calculated using the Stokes–Einstein equation for a spherical particle

$$r = \frac{k_B T}{6\pi\eta D} \quad (\text{Eq. R2-4})$$

where k_B is Boltzmann's constant, T the absolute temperature, and η the fluid viscosity. This procedure transforms particle trajectories into estimates of exosome size.^{5,6}

Revision 2-6

We appreciate this comment to clarify and enhance the manuscript. We have revised the manuscript accordingly to address the reviewer's concerns in the Supplementary Information.

Supplementary Information:

“**Supplementary Section 14.** Sample trajectories of exosomes with different sizes.

To perform mean-squared displacement (MSD) analysis, particle trajectories $r(t) = (x(t), y(t))$ are constructed by recording positions across N frames with an exposure time of Δt . At each time lag τ , the MSD is calculated by averaging the squared displacements between all position pairs separated by this specific lag interval as

$$MSD(\tau) = \langle |r(t + \tau) - r(t)|^2 \rangle \quad (\text{Eq. S8})$$

where the angular brackets denote ensemble averaging over all applicable time points t . For exosome characterization, high-speed imaging is conducted with an exposure time of 487.218 μs to capture rapid Brownian motion, with individual exosome trajectories extracted as shown in Supplementary Fig.11. When analyzing MSD as a function of time lag τ , freely diffusing particles exhibiting Brownian motion demonstrate a linear relationship described by

$$MSD(\tau) = error + b\tau \quad (\text{Eq. S9})$$

where the intercept term corrects for localization errors and the slope directly scales with the diffusion coefficient. In the case of three-dimensional Brownian motion projected onto a two-dimensional imaging plane, the coefficient is given by $D = b/4$. Finally, the exosome's radius r is calculated using the Stokes–Einstein equation for a spherical particle

$$r = \frac{k_B T}{6\pi\eta D} \quad (\text{Eq. S10})$$

where k_B is Boltzmann's constant, T the absolute temperature, and η the fluid viscosity. This procedure transforms particle trajectories into estimates of exosome size.^{4,5}”

7. On page 10, line 301, the manuscript states: “This leads to a higher dark mode background level as well as a degraded TP2 mode in silver meta-atoms, which is reflected in the broader wavelength response shown in Fig. 4i compared to the ideal meta-atom resonance.” However, Fig. 4i presents the scattering enhancement factor over the visible range, not a resonance broadening effect. This suggests that the figure reference may be incorrect. The authors should verify the figure labeling

and clarify which figure actually demonstrates the resonance broadening due to fabrication imperfections— perhaps Fig.2e?

Revision 2-7

Thank you for your detailed review. We find the reviewer's comment is correct. We have updated the manuscript to clarify this.

Main text:

“... This leads to a higher dark mode background level as well as a degraded TP2 mode in silver meta-atoms, which is reflected in the broader wavelength response shown in Fig. 2e compared to the ideal meta-atom resonance. ...”

Overall, the manuscript presents a novel and compelling approach with clear potential impact. I would strongly support its publication once the above clarifications and minor revisions are addressed.

In conclusion, we sincerely thank the reviewer for the valuable comments, which have greatly helped improve our manuscript.

Reviewer #3 (Remarks to the Author):

Review Report:

The article titled, “Meta-amplified dark-field interferometric scattering microscopy” by Lee et al proposes a technique (MAD-iSCAT) that combines dark field microscopy and interference to detect nanoparticles on a plasmonic metasurface. In this report, the authors demonstrated the technique by detecting polymer-nanoparticles and exosomes. They claimed the method to be better than the existing iSCAT technique in terms of sensitivity and contrast. I noted that the first three figures are largely schematic and simulation. The actual results are mainly limited to Fig. 4 and Fig. 5. Moreover, the results are shown on nanoparticles, rather than real systems such as cells, making the work somewhat limited in scope. Here are my detailed comments :

1. One of the major concerns is the lack of validation and poor understanding of what transpires at the metasurface related to field-particle interaction.

We thank the reviewer for the valuable comment. We acknowledge that clear theoretical evidence was not provided in the original manuscript. To address this point, we have developed a more comprehensive theoretical framework to elucidate the underlying mechanism of MAD-iSCAT. As this issue was also raised by other reviewers and discussed in detail over four pages, we kindly refer the reviewer to our previous response in the section **Theoretical framework for contrast and SNR enhancement in MAD-iSCAT** on page 5. To summarize, the underlying working principles of MAD-iSCAT have now been thoroughly investigated to explain the enhancement of contrast and SNR in iSCAT. We hope this satisfactorily addresses the reviewer’s concerns.

Revision 3-1

We have incorporated a comprehensive theoretical discussion and comparison of imaging contrast and SNR with conventional iSCAT. The manuscript has been revised accordingly to the reviewer’s concern. Please refer to **Revision 1-2** for detailed updates.

2. The particles used by authors are comparatively large. The technique (MAD-iSCAT) needs to demonstrate for small particle (<10 nm nanoparticles and < 300 kDa proteins) detection. This will strengthen the claim experimentally that MAD-iSCAT is better than iSCAT.

We thank the reviewer for this insightful comment. We respectfully argue that our work demonstrates significantly enhanced sensitivity and contrast compared to conventional iSCAT imaging.

To rigorously compare detection sensitivity across different iSCAT technologies, we employ a standardized metric that accounts for varying experimental conditions. In scattering-based detection, the measured signal from a given particle scales linearly with total photon dosage, the product of effective exposure time (EET) and incident power density P_D . Assuming a shot noise-limited system, the square root of the EET and P_D product $(\text{EET} \times P_D)^{0.5}$, enables a relative comparison of the detection SNR.

The theoretical study (Becker et al., ACS Photonics 10, 2699-2710 (2023)) predicts that conventional iSCAT on bare glass can achieve SNR ~ 21 for 66 kDa proteins under optimized conditions (illumination P_D of 0.1 MW/cm², EET = 100 ms, and $\lambda = 445$ nm). Many prior experimental studies do not report image SNRs for iSCAT but plasmonic scattering microscopy (PSM), where a single rough metallic film is used (Zhang et al., Nat. Methods 17, 1010-1017 (2020)), reaches an image SNR of ~ 11 for 385 kDa IgA protein with EET = 100 ms. The comparison is summarized below.

Table R3-1. A comparison of MAD-iSCAT against conventional iSCAT and plasmonic scattering microscopy (PSM, plasmonic iSCAT).

Method	Targets	EET (ms)	λ (nm)	P_D (W/cm ²)	$(\text{EET} \times P_D)^{0.5}$ $((\text{W} \cdot \text{ms}/\text{cm}^2)^{0.5})$	SNR	Reference
MAD-iSCAT	440 kDa	1 × 2	450	42	9.2	9	This study
Theoretical study (iSCAT)	66 kDa	100	445	100,000	3,162	21	ACS Photonics 10 , 2699-2710 (2023)
PSM	385 kDa	50 × 2	670	2,000	447	11	Nat. Methods 17 , 1010-1017 (2020)

In comparison, MAD-iSCAT achieves SNR ~ 9 for 440 kDa ferritin under dramatically reduced photon dosage. By normalizing the SNR values by target molecular weight, $(\text{EET} \times P_D)^{0.5}$, and λ^4 , the projected SNR for theoretical iSCAT and PSM are 0.91 and 1.55 under the same illumination conditions of MAD-iSCAT. Our approach demonstrates an **SNR enhancement factor of 23.2** relative to conventional iSCAT and a **7.1-fold improvement over PSM**. While PSM's enhancement originates solely from the field enhancement provided by the uniform gold film, MAD-iSCAT's superior performance stems from the perturbed bright modes of the plasmonic metasurface, consistent with our theoretical predictions.

As noted by the reviewer, further measurements with smaller molecules would robustly establish MAD-iSCAT's advantage. This measurement was an objective emphasized by two reviewers in the initial review round, and we accomplished the 20-nm polystyrene beads and ferritin measurement. Our current experimental configuration, based on a supercontinuum laser with narrow transmission linewidth, already operates at maximum available laser power. Theoretical analysis (Becker et al., ACS Photonics 10, 2699–2710 (2023)) indicates that the detection threshold of SNR = 3 in our current system

with SNR enhancement corresponds to a molecular weight of approximately 141 kDa (representing a substantial improvement compared to required 3,279 kDa without SNR enhancement). Extending detection to even smaller proteins below this threshold would require higher-power illumination sources, which we plan to implement in future work.

Through both theoretical analysis (revised theoretical framework) and experimental validation (SNR comparison presented above), we have demonstrated that MAD-iSCAT achieves substantial SNR enhancement over existing technologies. We hope this comprehensive comparison adequately addresses the reviewer's concerns regarding sensitivity and future prospects for smaller particle detection.

Revision 3-2

To demonstrate the superior performance of MAD-iSCAT, we have revised the main text and expanded comparisons in the Supplementary Information.

Main text:

“... The image SNR shows that our approach demonstrates an SNR enhancement factor of 23.2 relative to conventional iSCAT (see details in Supplementary Section 13).”

Supplementary Information:

“**Supplementary Section 13.** The comparison of MAD-iSCAT and iSCAT.

Supplementary Table 2. Imaging performances of different interferometric scattering imaging techniques. (EET: effective exposure time, PSNR: peak signal-to-noise ratio, and P_D : incident power density).

Method	Targets	EET (ms)	λ (nm)	P_D (W/cm ²)	$(EET \times P_D)^{0.5}$ ((W·ms/cm ²) ^{0.5})	PSNR	Reference
MAD-iSCAT	20-nm PS bead and 440 kDa	1 × 2	450	42	9.2	51 (PS bead) 9 (protein)	This study
Theoretical study (iSCAT)	66 kDa	100	445	100,000	3,162	21	ACS Photonics 10 , 2699-2710 (2023)
Photonic resonator interferometric scattering microscopy	440 kDa	16 × 2	633	25	28.3	N/A	Nat. Commun. 12 , 1744 (2021)
Quantitative mass imaging	53–803 kDa	300 × 2	445	420,000	15,874.5	N/A	Science 360 , 423-427 (2018)

Single-protein optical holography	90-540 kDa	6.25×2	450	50,000	790.6	N/A	Nat. Photon. 18 , 388 (2024)
Evanescent scattering microscopy	28-nm PS bead and 66 kDa	20×2	450	60,000	1,549.2	N/A	Nat. Commun. 13 , 2298 (2022)
Plasmonic scattering microscopy	26-nm PS bead and 385 kDa	50×2	670	200 (PS bead) 2,000 (protein)	141.4 (PS bead) 447.2 (Protein)	11 (protein)	Nat. Methods 17 , 1010-1017 (2020)

We compare MAD-iSCAT with other advanced interferometric scattering-based imaging techniques. ... Although these comparisons are indirect (since variations in imaging components can affect η and PSNRs are not well reported), Supplementary Table 2 summarizes the reported performance metrics. The exposure time is calculated from their frame rates. By normalizing the PSNR values by target molecular weight, $(EET \times P_D)^{0.5}$, and λ^4 , the projected PSNR for theoretical iSCAT and plasmonic scattering microscopy are 0.91 and 1.55 under the same illumination conditions of MAD-iSCAT. Our approach demonstrates an SNR enhancement factor of 23.2 relative to conventional iSCAT and a 7.1-fold improvement over plasmonic scattering microscopy. Beyond this enhanced sensitivity, MAD-iSCAT achieves the highest imaging throughput reported for interferometric scattering microscopy. This performance derives from our single-frame differential imaging approach, where acquisition speed is limited only by the camera readout rate rather than requiring temporal averaging or post-processing integration, while detection sensitivity can be further enhanced by increasing laser power.”

3. As claimed, high throughput and sensitivity needs further justification.

As discussed in **Revision 3-2**, the scattering enhancement mechanism of MAD-iSCAT is strong enough that we do not need to perform data averaging to achieve a good SNR and therefore leads to more than one order imaging speed improvement compared to other iSCAT methods. The scaling of the metasurface has also been considered during the design process for future large field-of-view high-throughput imaging. Should a larger patterned area be required, only one master mold is needed from e-beam lithography, and the metasurfaces can be mass-produced through nanoimprinting.

In addition to the reduced data acquisition time, we would also like to point out the importance of the reduced incident light intensity requirement. For practical biological imaging, especially in a cellular environment, a lower illumination intensity is preferred to avoid photodamage to the sample. The required intensity of MAD-iSCAT is less than that of a typical fluorescence wide-field microscope (100 W/cm^2), providing access to prolonged high-throughput bioimaging.

Revision 3-3

We appreciate the reviewer for the valuable comments. Please refer to **Revision 3-2**.

4. From the present results, the proposed technique does not appear to offer significant advance compared to iSCAT.

We thank the reviewer for this comment. We respectfully argue that our responses to the reviewer's **Comments 1 and 2** provide comprehensive evidence of MAD-iSCAT's significant advancement over conventional iSCAT, including: (1) a detailed theoretical framework with equations describing the SNR and contrast enhancement mechanisms, and (2) experimental validation demonstrating a 23.2-fold SNR enhancement compared to conventional iSCAT when normalized for experimental conditions. Please refer to our responses to the reviewer's **Comments 1 and 2** for the complete analysis.

Also, in general, we envision the unique scattering imaging mechanism of MAD-iSCAT will represent the future of state-of-the-art iSCAT. With conventional iSCAT techniques, particle sensitivity improvement has plateaued as we are reaching the practical limit of imaging dynamic range and shot noise limit using existing hardware. The concept of using super-scattering meta-atom mode and switching between dark-mode and bright-mode of a nanoantenna array opens up a new degree of freedom in designing a scattering imaging system.

Revision 3-4

We appreciate the reviewer for the comments. We have updated the discussion section to emphasize the importance of MAD-iSCAT.

Main text:

“... It is important to note that the current MAD-iSCAT has a detection volume more constrained than conventional iSCAT and suffers from fabrication imperfections, array inhomogeneities, and finite-size effects. Nonetheless, MAD-iSCAT combines the advantages of iSCAT and meta-atom arrays, opening new opportunities for optimization in label-free sensing applications. MAD-iSCAT has the potential to dramatically improve single-molecule mass imaging throughput by enhanced fabrication approaches, such as self-assembly⁵³ and large array geometries for expanded field-of-view and suppressed background noise. ...”

5. The abstract lacks to mention key details / outputs of the MAD-iSCAT technique. This include, (i) radiation modes and why they are special, (2) contrast enhancement, and (3) the level of suppression of the background. These information should have been mentioned in the abstract.

Revision 3-5

We appreciate the reviewer for the comments. We have updated the abstract to emphasize the key aspects of MAD-iSCAT.

Abstract:

“... By employing a metasurface comprising sub-diffraction plasmonic meta-atom arrays, MAD-iSCAT generates bright radiation modes that intensely scatter light toward the far field in the presence of a detection nanoparticle, substantially amplifying the sensitivity. In the absence of a nanoparticle, the metasurface produces minimal background due to the dark collective mode, resulting in improved image contrast. ...”

6. The manuscript lacks schematic of detailed optical setup in the main manuscript. This is essential to understand the proposed technique, instrumentation, and system development.

We appreciate the reviewer for the comments. We have added the schematics of the optical setup in the main text.

Revision 3-6

Main text:

Fig. 5 MAD-iSCAT imaging in aqueous environments. a, A schematic of the optical setup for the MAD-iSCAT in aqueous environments (A: axicon, L: lens, BM: beam block, BS: beam splitter, and BFP: back-focal plane). The right inset represents the measured BFP image of the incident beam and the detection NA. Solid arrows along the yellow beam path indicate the incident and reflected beams, while the dashed arrow represents the scattered light. ...”

We have revised the manuscript accordingly to address the reviewer's concerns.

Methods:

“... The dark-field ring at the back focal plane of the water-immersion objective is generated and its size adjusted using an axicon and two lenses in series, as illustrated in Fig. 5a. An example is shown in the inset of Fig. 5a. ... Incident angles are selected to satisfy the same plasmon modes excitation with $k_x = k_0 \sin 75^\circ = k_{\text{water}} \sin 47^\circ = k_{\text{PMMAS}} \sin 40^\circ$”

7. The authors mention, “By careful engineering meta-atom arrays, the presence of nanoscatterer.” The statement is vague, needs elaborate explanation, and must be substantiated.

Revision 3-7

We appreciate the reviewer for the comments. The detailed design of the metasurface is described in the metasurface characterization section of the main text. We have updated the manuscript to avoid confusion.

Main text:

“... When a nanoscatterer approaches, it disturbs the local resonance mode of the meta-atom elements and alters the phase profile of the sub-diffraction grating (Fig. 1f). The design of the meta-atom is chosen at a super-scattering resonance mode, such that the amplitude A_{ij} and phase ϕ_{ij} of the resonance mode are sensitive to the nearby environmental change due to the presence of the nanoscatterer, and the perturbed mode within the sub-diffraction array radiates towards the detection optics. ...”

8. In the results section, the authors claim high contrast. This is not substantiated and far from what is evident from the images shown in Fig. 4 and Fig. 5.

We thank the reviewer for the comments. We would like to argue that our method in fact produces scattering images with excellent contrast compared to conventional iSCAT, as shown in Fig. 5. For PS beads with sizes of 22, 41, and 81 nm, regular iSCAT yields image contrasts of $0.6 \pm 0.1\%$, $3.9 \pm 1.7\%$, and $23.9 \pm 3.1\%$ ($n = 127, 137,$ and 38), respectively, with 116 ms EET. In contrast, MAD-iSCAT demonstrates image contrasts of $14.9 \pm 4.5\%$, $156.5 \pm 46.6\%$, and $869.1 \pm 254.6\%$ ($n = 451, 246,$ and 20) for the same particle sizes. MAD-iSCAT shows an enhancement factor of 26 ± 7.5 for the 22 nm beads compared to regular iSCAT, employing a single-frame differential process with 2 ms EET.

To visually demonstrate the enhanced contrast from MAD-iSCAT, we have included the comparison of conventional iSCAT with MAD-iSCAT in the revised figure with a corresponding color bar. The contrast improvement can be clearly seen in the color bar scale.

9. One of the shortcomings of the present paper is the lack of experimental validation and application on real systems such as cell. Most of the results appear to be simulation based (Fig. 1-3). Actual experimental results are largely limited to Fig. 4 and Fig. 5. It should be other way round i.e., experimental validation must have a larger footprint.

We thank the reviewer for the comments. MAD-iSCAT provides a new scattering imaging mechanism that has not been explored before. It is crucial to present new formulations that are required to describe the physical process. Therefore, a significant portion of the manuscript is devoted to the theoretical analysis and numerical calculation of the proposed technique.

Revision 3-8

We appreciate the reviewer for the comments. To address the reviewer's concern, we have significantly revised the figure layout and content. Specifically, we have streamlined and reduced Fig. 3, which contains numerical calculations. Also, we have rearranged the main texts and experimental figures (now Fig. 4, 5, 6), and expanded their corresponding descriptions to provide a more detailed and larger footprint for the experimental validations and applications.

Main text:

“... We also conduct MAD-iSCAT imaging of small nanoparticles in aqueous environments, as shown in Fig. 5a. The imaging system is constructed with a custom axicon-based water-immersion dark-field microscopy.⁴⁸ Fig. 5 b,c compares MAD-iSCAT and regular iSCAT using PS nanobeads with diameters of 22, 41, and 81 nm. ...”

10. In Fig. 5, direct comparison of MAD-iSCAT and iSCAT image is missing.

We thank the reviewer for the suggestion. We have demonstrated ferritin MAD-iSCAT imaging with a significantly higher contrast of 0.027 compared to the literature value of 0.001 using conventional iSCAT.⁷ The 27-fold contrast improvement is in line with our demonstration of 25-fold contrast improvement using 20 nm PS beads shown in Fig. 5. In addition, it is noteworthy that the reference paper has a much higher illumination requirement of 3381, $(EET \times P_D)^{0.5}$ product (EET: effective exposure time and P_D : power density). In contrast, MAD-iSCAT requires only 9.2, $(EET \times P_D)^{0.5}$ product to obtain the 27-fold stronger contrast image.

11. There are many such points that needs to be take care-off in the manuscript. While the technique appears to be somewhat new, but it does little to advance the field and lacks experimental validation & application. This is in addition to the detailed comments by other two reviewers. The investigation appears to be in its initial stages and requires a lot more conclusive validation to show that MAD-iSCAT is better than iSCAT. Specifically, the images are not convincing and appears to be mild enhancement (visually). Overall, the work may be premature for publication at this stage.

We appreciate the reviewers' detailed feedback. To summarize this study, we want to emphasize the following points.

- (1) Conventional iSCAT sensitivity is constrained by shot noise and the practical limits of imaging dynamic range, requiring either high illumination intensities (risking photodamage to biological samples) or extensive temporal averaging. MAD-iSCAT overcomes these limitations by leveraging super-scattering meta-atom modes and switching between dark and bright modes of a meta-atom array. This introduces additional interference terms with enhanced scattering fields, amplifying detection signals without increasing photon flux or integration times. As mentioned in the response, the new contrast and SNR of MAD-iSCAT are represented as

$$C_{MAD-iSCAT} = \frac{|E_{BM}|^2 + |E_s|^2 + 2|E_{BM}||E_s|\cos\beta_1 + 2|E_s||E_{BG}|\cos\beta_2 + 2|E_{BG}||E_{BM}|\cos\beta_3}{|E_{BG}|^2}$$

$$\approx C_{iSCAT} + \frac{|E_{BM}|^2 + 2|E_{BM}||E_s|\cos\beta_1 + 2|E_{BG}||E_{BM}|\cos\beta_3}{|E_{BG}|^2}$$

and

$$SNR_{MAD-iSCAT} = \frac{|E_{BM}|^2 + |E_s|^2 + 2|E_{BM}||E_s|\cos\beta_1 + 2|E_s||E_{BG}|\cos\beta_2 + 2|E_{BG}||E_{BM}|\cos\beta_3}{|E_{BG}|}$$

$$\approx SNR_{iSCAT} + \frac{|E_{BM}|^2 + 2|E_{BM}||E_s|\cos\beta_1 + 2|E_{BG}||E_{BM}|\cos\beta_3}{|E_{BG}|}$$

respectively. This approach marks a leap beyond incremental improvements, enabling enhanced label-free scattering detection.

- (2) MAD-iSCAT is the first demonstration of exploiting both amplitude and phase responses of the metasurface's collective modes for far-field scattering detection. Unlike many other near-field techniques, such as surface-enhanced Raman scattering or plasmonic hot-spot-based imaging, MAD-iSCAT utilizes the radiative properties of meta-atom arrays to detect nanoparticles. We demonstrate this concept with a robust theoretical framework, supported by numerical simulations and experiments using various nanoparticles, including polystyrene beads, exosomes, and protein molecules. **Experimentally, we have shown 23.2-fold SNR enhancement compared to previous iSCAT studies.** These results pave the way for a new paradigm in scattering detection and optimized metasurface-based particle detection.

Revision 3-9

To address the reviewer's concern, in addition to **Revision 3-1 and 3-2**, we have revised the discussion section.

Main text:

“... MAD-iSCAT underscores that the enhanced image contrast arises not only from the field enhancement effects near the silver surfaces but also from the additional interference with the radiation modes of silver meta-atom arrays. ...”

In conclusion, we sincerely thank the reviewer for the valuable feedback, which has greatly strengthened the manuscript. We believe these revisions address the concerns raised and highlight the transformative potential of MAD-iSCAT in advancing scattering-based imaging.

Reviewer #4 (Remarks to the Author):

This manuscript reports the use of a plasmonic metasurface to perform interferometric scattering microscopy (iSCAT). The authors claim that the radiation mode supported by the metasurface interferes with the scattered light from nanoparticles, thereby producing high-contrast interferometric images. Reviewer #1 and Reviewer #2, both experts in iSCAT, have already raised critical questions regarding the novelty and performance of the work, to which the authors have responded. As a specialist in plasmonics and metasurfaces, I will not repeat the iSCAT-related issues already discussed, but instead provide three comments from the perspective of employing metasurfaces for iSCAT.

1. The primary novelty of this work lies in the conceptual innovation of employing a metasurface for iSCAT for the first time. However, the manuscript lacks clarity in its presentation and fails to provide a concise physical picture. As a result, the authors introduce a section, "Principles of contrast amplification in interferometric images," supported by eight equations; this style is more reminiscent of Physical Review rather than Nature Communications. Similarly, the section "The formation of scattering image by metasurface" relies heavily on numerical simulations to justify the mechanism behind, which is not a particularly strong form of evidence.

We thank the reviewer for this constructive feedback regarding the clarity and presentation of our theoretical framework. In response to these concerns, we have substantially revised the "Principles of contrast amplification in interferometric images" section. We have also developed a more comprehensive theoretical framework to elucidate the underlying mechanism of MAD-iSCAT. As this issue was also raised by other reviewers and discussed in detail over four pages, we kindly refer the reviewer to our previous response in the section **Theoretical framework for contrast and SNR enhancement in MAD-iSCAT** on page 5.

Revision 4-1

The manuscript has been revised accordingly to the reviewer's concern. Please refer to **Revision 1-2** for detailed updates. We have also streamlined Fig. 3 related to "The formation of a scattering image by a metasurface" section to present a more concise visual representation that emphasizes the essential physical mechanisms.

Main text:

2. It is well established that plasmonic metasurfaces can enhance light–matter interactions and consequently boost scattering signals. Therefore, the reported contrast enhancement by one to two orders of magnitude (Fig. 4) is in fact expected. However, such enhancement comes at the cost of local heating due to metallic nanostructures. This heating may severely limit the applicability of iSCAT for biologically relevant tasks such as single-molecule tracking. The manuscript does not discuss this critical issue, and such a discussion should be included.

We appreciate this observation. Thank you for highlighting this important factor. We used a relatively low laser power density of 42 W/cm². In previous studies of silver nanospheres (10–50 nm radius) at 4,000 W/cm², where scattering cross-sections peaked as 1×10⁻¹⁴ m², temperature rises of 0.5–3 °C were confirmed. Since our SSP mode features cross-sections in the same order of magnitude but operates at 100× lower power density, we estimate a temperature increase of less than 1 °C. Under these conditions, biological samples are not affected by plasmonic heating, alleviating concerns about thermal damage.⁴

Revision 4-2

We appreciate this observation. We have revised the manuscript accordingly to address the reviewer's concerns.

Main text:

“... The laser power employed is sufficiently low to avoid significant photothermal effects and our protective layer also minimizes the heating effect on detection targets and prevents photodamage to silver meta-atoms.⁵¹ ...”

3. Comparative analysis is essential to place the method into context. For example, Table R1-2 in the response letter provides a very useful benchmark. I strongly recommend that the authors incorporate this table into the main text and expand the discussion accordingly.

Revision 4-3

We appreciate this observation. We have revised the manuscript accordingly to address the reviewer's concerns.

Main text:

“... MAD-iSCAT has the potential to dramatically improve single-molecule mass imaging throughput by orders of magnitude through enhanced fabrication approaches such as self-assembly for better metasurface quality⁵³ and large array geometries to enable high-throughput nanoparticle imaging with expanded field-of-view and suppressed background noise. ...”

Supplementary information:

“Supplementary Section 13. The comparison of MAD-iSCAT and iSCAT.

Supplementary Table 2. Imaging performances of different interferometric scattering imaging techniques. (EET: effective exposure time, PSNR: peak signal-to-noise ratio, and P_D : incident power density).

Method	Targets	EET (ms)	λ (nm)	P_D (W/cm ²)	$(EET \times P_D)^{0.5}$ ((W·ms/cm ²) ^{0.5})	PSNR	Reference
MAD-iSCAT	20-nm PS bead and 440 kDa	1 × 2	450	42	9.2	51 (PS bead) 9 (protein)	This study
Theoretical study (iSCAT)	66 kDa	100	445	100,000	3,162	21	ACS Photonics 10 , 2699-2710 (2023)
Photonic resonator interferometric scattering microscopy	440 kDa	16 × 2	633	25	28.3	N/A	Nat. Commun. 12 , 1744 (2021)

Quantitative mass imaging	53–803 kDa	300 × 2	445	420,000	15,874.5	N/A	Science 360 , 423-427 (2018)
Single-protein optical holography	90-540 kDa	6.25 × 2	450	50,000	790.6	N/A	Nat. Photon. 18 , 388 (2024)
Evanescence scattering microscopy	28-nm PS bead and 66 kDa	20 × 2	450	60,000	1,549.2	N/A	Nat. Commun. 13 , 2298 (2022)
Plasmonic scattering microscopy	26-nm PS bead and 385 kDa	50 × 2	670	200 (PS bead) 2,000 (protein)	141.4 (PS bead) 447.2 (Protein)	11 (protein)	Nat. Methods 17 , 1010-1017 (2020)

We compare MAD-iSCAT with other advanced interferometric scattering-based imaging techniques. ... Although these comparisons are indirect (since variations in imaging components can affect η and PSNRs are not well reported), Supplementary Table 2 summarizes the reported performance metrics. The exposure time is calculated from their frame rates. By normalizing the PSNR values by target molecular weight, $(EET \times P_D)^{0.5}$, and λ^4 , the projected PSNR for theoretical iSCAT and plasmonic scattering microscopy are 0.91 and 1.55 under the same illumination conditions of MAD-iSCAT. Our approach demonstrates an SNR enhancement factor of 23.2 relative to conventional iSCAT and a 7.1-fold improvement over plasmonic scattering microscopy. Beyond this enhanced sensitivity, MAD-iSCAT achieves the highest imaging throughput reported for interferometric scattering microscopy. This performance derives from our single-frame differential imaging approach, where acquisition speed is limited only by the camera readout rate rather than requiring temporal averaging or post-processing integration, while detection sensitivity can be further enhanced by increasing laser power.”

In summary, the manuscript presents an interesting concept and demonstrates promising experimental results, but the novelty and implications are not clearly articulated. I recommend a major revision before the work can be considered further.

In conclusion, we sincerely appreciate the reviewer’s insightful feedback, which has significantly enhanced the quality of our manuscript.

References

1. Zhang, P. F. et al. Plasmonic scattering imaging of single proteins and binding kinetics. *Nat. Methods* **17**, 1010-1017 (2020).
2. Zhang, P. F. et al. Evanescent scattering imaging of single protein binding kinetics and DNA conformation changes. *Nat. Commun.* **13**, 2298 (2022).
3. Tanaka, Y., Kaneda, S. & Sasaki, K. Nanostructured potential of optical trapping using a plasmonic nanoblock pair. *Nano Lett.* **13**, 2146-2150 (2013).
4. Kailash & Verma, S. S. Exploring thermoplasmonic profiles of some noble metallic nanospheres. *Mater. Today Commun.* **33**, 104795 (2022).
5. Michalet, X. Mean square displacement analysis of single-particle trajectories with localization error: Brownian motion in an isotropic medium. *Phys. Rev. E Stat. Nonlin. Soft Matter Phys.* **82**, 041914 (2010).
6. Qian, H., Sheetz, M. P. & Elson, E. L. Single particle tracking. Analysis of diffusion and flow in two-dimensional systems. *Biophys J.* **60**, 910-921 (1991).
7. Piliarik, M. & Sandoghdar, V. Direct optical sensing of single unlabelled proteins and super-resolution imaging of their binding sites. *Nat. Commun.* **5**, 4495 (2014).

Reviewer #1 (Remarks to the Author):

Thank you for addressing all the comments satisfactorily.

Response: We sincerely thank the reviewer for the positive evaluation. We deeply appreciate the reviewer's time and constructive input, which helped us improve the manuscript.

Reviewer #2 (Remarks to the Author):

The manuscript has been significantly improved, and the authors have clearly addressed all of the points I originally raised. I believe it is now of sufficient quality for publication in Nature Communications.

Response: We sincerely appreciate the insightful feedback provided, which contributed significantly to refining the quality of our work. We thank the reviewer for the positive recommendation.

Reviewer #4 (Remarks to the Author):

The authors responded very positively and effectively to the issues I mentioned earlier. The current version is satisfactory, and I recommend publication.

Response: We sincerely thank the reviewer for the thoughtful feedback that guided these revisions. We thank the reviewer for the positive recommendation.

We would like to express our sincere gratitude to all reviewers and the editorial office for their time and insightful feedback, which greatly improved the quality of our manuscript.

In addition, we also fixed a mistake in one of the figures and corrected a few numbers accordingly.